# HELENE: Hessian Layer-wise Clipping and Gradient Annealing for Accelerating Fine-Tuning LLM with Zeroth-Order Optimization

## Abstract

Fine-tuning large language models (LLMs) poses significant memory challenges, as the back-propagation process demands extensive resources, especially with growing model sizes. Recent work, MeZO, addresses this issue using a zeroth-order (ZO) optimization method, which reduces memory consumption by matching the usage to the inference phase. However, MeZO experiences slow convergence due to varying curvatures across model parameters. To overcome this limitation, we introduce HELENE, a novel scalable and memory-efficient optimizer that integrates annealed A-GNB gradients with a diagonal Hessian estimation and layer-wise clipping, serving as a second-order pre-conditioner. This combination allows for faster and more stable convergence. Our theoretical analysis demonstrates that HELENE improves convergence rates, particularly for models with heterogeneous layer dimensions, by reducing the dependency on the total parameter space dimension. Instead, the method scales with the largest layer dimension, making it highly suitable for modern LLM architectures. Experimental results on RoBERTa-large and OPT-1.3B across multiple tasks show that HELENE achieves up to a $20\times$ speedup compared to MeZO, with average accuracy improvements of 1.5%. Furthermore, HELENE remains compatible with both full parameter tuning and parameter-efficient fine-tuning (PEFT), outperforming several state-of-the-art optimizers. The codes will be released after reviewing.

## 1 Introduction

LLMs have demonstrated remarkable capabilities across various downstream tasks. Fine-tuning these models has become the standard approach for improving task-specific performance, in which the first-order optimizers like Stochastic Gradient Descent (SGD) (Robbins & Monro, 1951), Adam (Diederik, 2014) and AdamW (Hutter & Loshchilov, 2017) are widely used. While effective, however, these methods demand significant memory resources primarily due to the backpropagation process, which makes fine-tuning challenging, especially for large-scale models. To overcome this limitation, Malladi et al. (2023) proposed a memory-efficient zeroth-order optimizer (MeZO) that estimates gradients using only two forward passes per training step, contributing to considerable memory savings.

However, recent studies show that loss functions in deep learning often exhibit heterogeneous curvatures across different model parameters and different model layers (Sagun et al., 2016; Ghorbani et al., 2019; Zhang et al., 2022; Yao et al., 2020), which poses challenges to zeroth-order (ZO) optimization. This variation in curvature can overall hinder training efficiency and lead to the sub-optimal solution. To address this issue, more advanced techniques are required, such as incorporating second-order information to better account for curvature differences (Liu et al., 2023; Tran & Cutkosky, 2022; Jahani et al., 2021). However, in ZO optimization, directly computing the Hessian from first-order derivatives is nearly impossible, and partial Hessian evaluations are computationally intensive, leading to slower convergence. Moreover, we also observe that these methods like Naive Newton's method and Sophia (Liu et al., 2023) fail in fine-tuning LLMs in practice as illustrated in Figure 1 and Figure 2.

To overcome the aforementioned challenges, we propose HELENE, a novel optimizer designed to estimate second-order curvature information efficiently in the context of ZO optimization. Originating

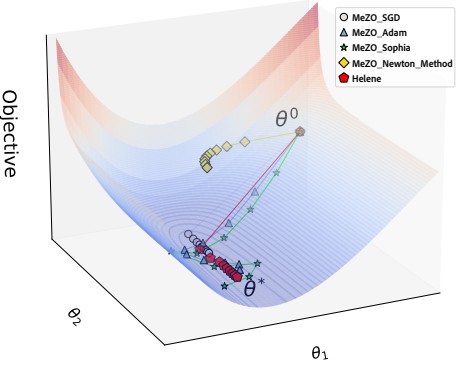

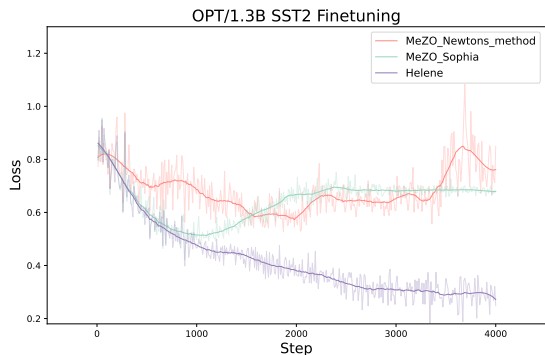

Figure 1: The motivating toy example. HE-LENE can maintain stable updates when facing curvature issues, while other second-order optimizers are severely affected by them.

Figure 2: Comparison of HELENE with Newton's method and Sophia. The performance of this training loss cross-checks with the toy sample in Figure 1.

from label-sampling-based Gaussian-Newton Garlett (GNB) Estimator (Schraudolph, 2002; Wei et al., 2020; Martens, 2020), in our HELENE algorithm, we introduce a label-sampling-free and efficient Hessian estimator called the asymptotic Gauss-Newton-Bartlett estimator (A-GNB) Estimator, which estimates the diagonal of the Hessian matrix. A-GNB is proven to asymptotically converge to the unbiased Gauss Newton matrix. Additionally, HELENE includes a layer-wise adaptive clipping mechanism that enables more precise curvature-aware updates, while magnitude-based clipping helps prevent the overestimation of extreme values in the Hessian diagonal. Unlike existing methods that rely on global clipping, which can distort gradient signals, HELENE preserves the integrity of gradient information by applying clipping on a per-layer basis.

One of the key innovations of HELENE is its ability to adaptively clip Hessian updates according to the curvature of each layer, which significantly enhances convergence rates. While the convergence of state-of-the-art optimizers like MeZO-Sophia which may require $\mathcal{O}(d)$ steps; in contrast, the convergence of HELENE requires significantly less steps, which is $\mathcal{O}(\max_i d_i)$ based on the largest layer dimension $\max_i d_i$ across all layers, making it more suitable for modern deep architectures. We observed that zero-order methods require a stronger emphasis on momentum due to the increasing noise in SPSA gradient estimation as optimization progresses, due to the greater difficulty of training as the model parameters approach a local minimum, causing the noise magnitude from sampling perturbations to exceed that of the true gradient signal.So contracting to most momentum design in first-order methods, the proposed HELENE algorithm also integrates an novel annealing exponential moving average (EMA) of the gradients, tailored for Zero-order methods, where the factor alpha dynamically reduce the weight of the gradient in the momentum update.

Overall, our key contributions can be summarized as follows:

1. HELENE integrates a novel asymptotic Gauss-Newton-Bartlett (A-GNB) estimator that efficiently estimates the diagonal of the Hessian matrix without the need for label sampling which may incur more noise in the Hessian estimation. This estimator asymptotically converges to the unbiased diagonal Gauss Newton matrix, improving the efficiency and precision of curvature-aware updates. In our proposed method, we also devise a new layer-wise adaptive clipping mechanism by adjusting Hessian updates according to the curvature of each layer. HELENE integrates an new annealing exponential moving average (EMA) of the gradients, ensuring robustness in non-convex loss landscapes.

2. Our theoretical analysis demonstrates that HELENE achieves improved convergence rates compared to existing methods, particularly for models with many layers. By reducing the convergence steps from $\mathcal{O}(d)$ to $\mathcal{O}(\max_i d_i)$, HELENE is provably more scalable for modern deep learning architectures, especially LLM fine-tuning.

3. HELENE achieves up to $20\times$ speedup compared to MeZO and improves performance by 1.5% on average. We conduct extensive experiments on RoBERTa-large and OPT-1.3B across various downstream tasks to verify HELENE's effectiveness. Furthermore, we demonstrate that HELENE not only remains compatible with full parameter tuning and PEFT, but also outperforms many of the latest optimizers across a range of tasks.

## 2 PRELIMINARIES

In this section, we briefly review essential background concepts related to zeroth-order optimization and diagonal Hessian approximation, which are fundamental to the design of our proposed method.

### 2.1 ZEROTH-ORDER GRADIENT ESTIMATORS AND MEZO

Zeroth-order (ZO) optimization has long been studied in the context of convex and non-convex objectives. One of the typical ZO gradient estimators is the simultaneous perturbation stochastic approximation (SPSA) (Spall, 1992; Maryak & Chin, 2001). Given a model with parameters $\boldsymbol{\theta} \in \mathbb{R}^d$ and loss function $\mathcal{L}$, SPSA estimates the gradient on a minibatch $\mathcal{B}$ as:

$$g_\epsilon(\boldsymbol{\theta}) = \frac{\mathcal{L}(\boldsymbol{\theta} + \epsilon \boldsymbol{z}; \mathcal{B}) - \mathcal{L}(\boldsymbol{\theta} - \epsilon \boldsymbol{z}; \mathcal{B})}{2\epsilon} \boldsymbol{z} \approx \boldsymbol{z}\boldsymbol{z}^\top \nabla \mathcal{L}(\boldsymbol{\theta}; \mathcal{B}) \tag{1}$$

where $\boldsymbol{z} \in \mathbb{R}^d$ with $\boldsymbol{z} \sim \mathcal{N}(\boldsymbol{0}, \boldsymbol{I}_d)$ and $\epsilon$ is the perturbation scale.

Building on the basic principles of ZO optimization, MeZO (Malladi et al., 2023) introduces a memory-efficient implementation of ZO-SGD. This approach reduces memory requirements, allowing optimization to proceed with the same memory usage as the inference phase of a model. The key innovation in MeZO lies in its use of a consistent random seed $s$ to sample the random vector $\boldsymbol{z}$, ensuring the same perturbation $\boldsymbol{z}$ at each step.

### 2.2 DIAGONAL HESSIAN APPROXIMATION

While zeroth-order methods like MeZO provide valuable tools for gradient estimation, optimization can be significantly enhanced by incorporating second-order information, such as curvature. However, directly computing and applying the full Hessian matrix is computationally expensive, particularly in high-dimensional parameter spaces. Specifically, directly applying the Hessian pre-conditioner by calculating the inverse Hessian and multiplying it with the gradient vector at each iteration $\boldsymbol{H}^{-1}\boldsymbol{g}$ is particularly computationally expensive. To address this challenge, inexact Newton methods have been developed, where approximations of the Hessian are used instead of the full matrix (Dembo et al., 1982; Bollapragada et al., 2019; Xu et al., 2020).

A simple yet effective alternative is to approximate the Hessian by its diagonal elements, which reduces computational complexity while retaining useful curvature information. In this approach, a general descent direction can be written as follows:

$$\Delta \boldsymbol{\theta} \approx \text{diag}(\boldsymbol{H})^{-1}\boldsymbol{g},$$

where $\text{diag}(\boldsymbol{H})$ represents the diagonal elements of the Hessian matrix. This method enhances optimization by enabling efficient inverse Hessian application and supporting inexact Newton methods, providing improved convergence in complex problems.

## 3 METHOD

In this section, we formally present HELENE in Section 3.2, with pseudo-code provided in Algorithm 1. In Section 3.4, we introduce A-GNB, followed by a detailed discussion of layer-wise clipped diagonal Hessian in Section 3.5.

### 3.1 MOTIVATION

**Highly variable curvature across different layers and parameters.** Fine-tuning large language models (LLMs) has become essential for achieving state-of-the-art performance on various downstream tasks. Commonly employed first-order optimizers such as Stochastic Gradient Descent

(SGD)(Robbins & Monro, 1951), Adam(Diederik, 2014), and AdamW (Hutter & Loshchilov, 2017) have proven effective in this regard. However, these methods require substantial memory, making them difficult to apply to large models in memory-constrained environments. To mitigate this, zeroth-order (ZO) optimizers, such as MeZO (Malladi et al., 2023), have been introduced, offering memory-efficient solutions by approximating gradients through forward passes. Nevertheless, even with memory savings, existing ZO methods encounter significant challenges when dealing with heterogeneous curvatures in LLMs, which can lead to inefficient convergence and sub-optimal solutions. One key challenge is the inability of optimizers to adapt to the highly variable curvature across different layers and parameters in large models (Sagun et al., 2016; Ghorbani et al., 2019; Zhang et al., 2022). While techniques that estimate second-order information—such as curvature-aware methods—have shown promise in improving optimization efficiency (Liu et al., 2023; Tran & Cutkosky, 2022; Jahani et al., 2021), they are challenging to integrate into ZO optimizers due to the noise from label-sampling and the difficulty of computing or approximating the Hessian efficiently in high-dimensional spaces.

**Limitation of EMA to balance short-term gradient noise and long-term convergence.** A commonly used technique to manage these curvature variations is the Exponential Moving Average (EMA), which smooths the gradient updates over iterations. However, EMA alone can be insufficient for highly non-convex loss landscapes, especially when it lacks mechanisms to adaptively adjust the weights between the past momentum and the current gradient in the presence of strong noise in gradient estimation. Without annealing, EMA risks accumulating excessive bias over time, particularly when ZO gradient estimation is noisy, leading to suboptimal convergence. This issue is compounded when the optimizer needs to balance short-term gradient noise and long-term convergence, calling for more ZO-specific strategies to mitigate these effects.

**Challenge in managing extreme curvature values using Universal clipping.** Furthermore, clipping the Hessian to manage extreme curvature values is another widely adopted strategy. Sophia (Liu et al., 2023), for example, performs global clipping with value 1 of Hessian-based updates to ensure numerical stability, which essentially can slow down the convergence. While effective at curbing extreme updates, applying a universal clipping threshold across all parameters is inherently suboptimal for models with heterogeneous curvatures. A universal clip might suppress meaningful gradient information in some layers while insufficiently addressing extreme Hessian values in others, thus limiting the optimizer's ability to adaptively handle the diverse learning dynamics across layers (Tran & Cutkosky, 2022). This approach may result in slower convergence or failure to escape saddle points and local maxima, where more flexible, curvature-aware updates are required (Yao et al., 2020).

To overcome these limitations, HELENE addresses both the limitation of EMA and the issue of global Hessian clipping. We instantiate MeZO-Gradient Descent, MeZO-Adam, MeZO-Newton's method, MeZO-Sophia, and HELENE on a simplified 2D problem to illustrate the advantages HELENE, as shown in Figure 1. A visual comparison of the methods reveals that while MeZO-Adam and MeZO-Gradient Descent struggle to converge effectively, Newton's method and Sophia find it hard to maintain stability when facing heterogeneous curvature, whereas HELENE succeeds. Refer to Section 5 for a more comprehensive empirical analysis, including up to $20\times$ faster convergence and improved accuracy across various tasks and datasets.

### 3.2 HELENE: Hessian Layer-wise Clipping and Gradient Annealing

In HELENE, we introduce an annealing mechanism to mitigate bias in SPSA-estimated gradients, combined with a clipped diagonal Hessian pre-conditioner that adjusts parameter update step sizes based on layer-wise curvature. First, the gradient is calculated using the SPSA, while the diagonal Hessian is efficiently estimated by the proposed new A-GNB method, to eliminate the noise incured in sampling labels from the model output used in GNB and Sophia. At each iteration, SPSA provides an estimate $g_t$ using two forward passes with random perturbations, and A-GNB returns $h_t$, the diagonal Hessian of the mini-batch loss.

We apply an exponential moving average (EMA) to both the gradient and diagonal Hessian estimates to reduce noise and improve stability. To further enhance convergence, we apply layer-wise magnitude-based clipping to the diagonal Hessian, ensuring extreme values do not disproportionately affect parameter updates. We provide our pseudo code in Algorithm 1 and each module description in the following section in details.

### 3.3 EMA OF DIAGONAL HESSIAN ESTIMATES

When using a mini-batch to compute the local Hessian (curvature), the resulting estimates are often noisy. The Hessian diagonal can fluctuate significantly across different parameter dimensions of the problem. Inspired by the exponential moving average (EMA) of gradient moments in Adam, we apply a similar technique to reduce noise in the Hessian diagonal estimates over iterations. The updated Hessian diagonal is computed in the following:

$$\boldsymbol{h}_t = \beta_2 \boldsymbol{h}_{t-k} + (1 - \beta_2)\hat{\boldsymbol{h}}_t,$$

where $\boldsymbol{h}_t$ represents the denoised Hessian diagonal at iteration $t$ and $\hat{\boldsymbol{h}}_t$ is the current estimate of the diagonal at the $k$-th iteration.

---

**Algorithm 1** HELENE with Layer-wise Clipping

---

1: **Input:** Initial parameters $\boldsymbol{\theta}_1$, step budget $T$, learning rate schedule $\{\eta_t\}_{t=1}^T$, hyperparameters $\{\lambda_i\}, \gamma, \beta_1, \beta_2, \epsilon$.
2: Set $\boldsymbol{m}_0 = 0$, $h_0 = 0$
3: **for** $t = 1$ **to** $T$ **do**
4:     Estimate gradient $\boldsymbol{g}_t$ from $\nabla L_t(\boldsymbol{\theta}_t)$ obtained from Eq. (1).
5:     $\alpha = \text{Anneal}(t)$
6:     $\boldsymbol{m}_t = \beta_1 \boldsymbol{m}_{t-1} + \alpha \boldsymbol{g}_t$
7:     **if** $t \bmod k = 1$ **then**
8:         Compute diagonal Hessian estimator $\hat{\boldsymbol{h}}_t = \text{A-GNB}(\boldsymbol{\theta}_t)$
9:         $\boldsymbol{h}_t = \beta_2 \boldsymbol{h}_{t-k} + (1 - \beta_2)\hat{\boldsymbol{h}}_t$
10:     **else**
11:         $\boldsymbol{h}_t = \boldsymbol{h}_{t-1}$
12:     **end if**
13:     Apply weight decay: $\boldsymbol{\theta}_t = \boldsymbol{\theta}_t - \eta_t \epsilon \boldsymbol{\theta}_t$
14:     For each layer $i$, update: $\boldsymbol{\theta}_{t+1,i} = \boldsymbol{\theta}_{t,i} - \eta_t \cdot \frac{\boldsymbol{m}_{t,i}}{\gamma \cdot \max(\boldsymbol{h}_{t,i}, \lambda_i) + \epsilon}$
15: **end for**

---

1: **Subroutine** Anneal(t)

$$\alpha \leftarrow \beta_1 + (1 - \beta_1) \cdot \exp(-t/T) \tag{2}$$

---

### 3.3.1 ANNEALING MECHANISM

As illustrated in Figure 5, the native gradient EMA introduces bias, which adversely affects the training process and results in an increase in loss during the later stages. To mitigate these issues, we introduce a gradient annealing mechanism to work in tandem with EMA. This adaptive adjustment is crucial for ensuring that the model becomes less influenced by noisy or outdated gradients in later stages. We observe that, unlike first-order methods such as SGD, Zero-order methods require a stronger emphasis on momentum due to the increasing noise in SPSA gradient estimation as optimization progresses, as illustrated in Appendix Figure 7. To address this, we introduce a novel annealing strategy tailored for Zero-order methods, where the factor $\alpha$, dynamically adjusts the weight of the gradient in the momentum update. The increase in noise in SPSA is likely due to the greater difficulty of training as the model parameters approach a local minimum, causing the noise magnitude from sampling perturbations to exceed that of the true gradient signal. Notably, our annealing approach is simple to implement, requiring the tuning of only a single hyperparameter.

At each iteration, the annealing mechanism computes $\alpha$ using an exponential decay schedule in Eq. 2, where $T$ is a predefined hyperparameter controlling the annealing rate. The increase in noise in SPSA is likely due to the greater difficulty of training as the model parameters approach a local minimum, causing the noise magnitude from sampling perturbations to exceed that of the true gradient signal. To address this, as $t$ increases, $\alpha$ gradually decreases to reduce the impact of gradient on the update, mitigating the bias introduced by EMA. This ensures that, in the later stages of training, the model focuses more on stable gradient estimates and less on noisy or rapidly changing updates via SPSA estimated gradient. The annealing mechanism is incorporated into the EMA update rule as line 6 in Algorithm 1. Via dynamical $\alpha$ the annealing mechanism ensures that the optimizer can effectively balance short-term noise with long-term convergence.

## 3.4 ASYMPTOTIC GAUSS-NEWTON-BARTLETT (A-GNB) ESTIMATOR

The original GNB (Martens, 2020) estimator relies on sampled labels $\hat{y}_b$ drawn from the categorical distribution based on the model's output. However, this induces stochasticity due to label sampling, which could be problematic when label distributions are highly imbalanced, as is the case in large language model (LLM) training. We propose a new estimator, which we call the **Asymptotic Gauss-Newton-Bartlett (A-GNB) Estimator**, that replaces the sampled labels $\hat{y}_b$ with the true labels $y_b$ and asymptotically converges to the true diagonal of the Gauss-Newton matrix, which is a biased estimator for the diagonal of the Hessian as shown below:

$$\nabla_{\boldsymbol{\theta}}^2 L(\boldsymbol{\theta}) \approx J_{\boldsymbol{\theta}} f(\boldsymbol{\theta}, \boldsymbol{x}) \cdot S \cdot J_{\boldsymbol{\theta}} f(\boldsymbol{\theta}, \boldsymbol{x})^\top \tag{3}$$

where $J_{\boldsymbol{\theta}} f(\boldsymbol{\theta}, x)$ is the Jacobian of the model's output $f(\boldsymbol{\theta}, \boldsymbol{x})$ with respect to the parameters $\boldsymbol{\theta}$, and $S = \frac{\partial^2 L(t,y)}{\partial t^2}$ is the second-order derivative of the loss w.r.t. the logits $t = f(\boldsymbol{\theta}, \boldsymbol{x})$ and $y \sim p(y|\boldsymbol{x})$, which implies that $S = \mathbb{E}_{y \sim p(\boldsymbol{\theta}, \boldsymbol{x})} \left[ \frac{\partial^2 L(t,y)}{\partial t^2} \right]$ assuming that $L$ is the Cross-Entropy loss. Consequently, the diagonal of the Gauss-Newton matrix for the mini-batch loss is estimated as:

$$diag(J_{\boldsymbol{\theta}} f(\boldsymbol{\theta}, \boldsymbol{x}) \cdot S \cdot J_{\boldsymbol{\theta}} f(\boldsymbol{\theta}, \boldsymbol{x})^\top)) = \mathbb{E}_{y \sim p(y|\boldsymbol{x})} \left[ diag(J_{\boldsymbol{\theta}} f(\boldsymbol{\theta}, \boldsymbol{x}) \frac{\partial L(t,y)}{\partial t} \frac{\partial L(t,y)}{\partial t}^\top J_{\boldsymbol{\theta}} f(\boldsymbol{\theta}, \boldsymbol{x})^\top) \right]$$

$$= \mathbb{E}_{y \sim p(y|\boldsymbol{x})} \left[ diag(\nabla_{\boldsymbol{\theta}} L(f(\boldsymbol{\theta}, \boldsymbol{x}), y) \nabla_{\boldsymbol{\theta}} L(f(\boldsymbol{\theta}, \boldsymbol{x}), y)^\top) \right]$$

$$\approx \frac{1}{B} \sum_{b=1}^{B} [(g(\boldsymbol{\theta}, \boldsymbol{x}_b, y_b)] \odot [(g(\boldsymbol{\theta}, \boldsymbol{x}_b, y_b)]$$

where $diag(\cdot)$ represents the diagonal elements of a matrix, and $B$ denotes the batch size and $g$ is the estimated gradient from Eq. (1). In contrast to GNB estimator, which includes sampling label $\hat{y}$ from the logit probability output from the model, we replace it by $y_b$, the true label, thereby avoiding the need for post-output label sampling. By eliminating the stochasticity induced by sampled labels $\hat{y}$, we reduce the variance caused by sampling noise, and it is especially beneficial in imbalanced data scenarios, when samples from minor class is rarely selected unless sampling significantly many times.

The estimated gradient terms now correspond directly to the true labels, and their outer product sums up to the true Gauss-Newton approximation of the Hessian. As the batch size $B \to \infty$, the A-GNB estimator converges to the true Hessian's diagonal:

$$\lim_{B \to \infty} \frac{1}{B} \sum_{b=1}^{B} [g(\boldsymbol{\theta}, \boldsymbol{x}_b, y_b)] \odot [g(\boldsymbol{\theta}, \boldsymbol{x}_b, y_b)] = diag(J_{\boldsymbol{\theta}} f(\boldsymbol{\theta}, \boldsymbol{x}) \cdot S \cdot J_{\boldsymbol{\theta}} f(\boldsymbol{\theta}, \boldsymbol{x})^\top)$$

Therefore, The A-GNB estimator asymptotically converges to the true diagonal of the Gauss-Newton matrix as $B$ increases.

---

**Algorithm 2** Asymptotic Gauss-Newton-Bartlett (A-GNB)

---

1: Parameters: $\boldsymbol{\theta}$
2: Draw a mini-batch of the input $\{x_b\}_{b=1}^{B}$
3: Estimate diagnal Hessian matrix by $\boldsymbol{h} = \sum_{b=1}^{B} [g(\boldsymbol{\theta}, \boldsymbol{x}_b, y_b)] \odot [g(\boldsymbol{\theta}, \boldsymbol{x}_b, y_b)]$
4: return $\boldsymbol{h}$

---

## 3.5 LAYWERWISE CLIPPED DIAGONAL HESSIAN TO HELP NEWTON'S METHOD

As discussed in the motivating examples, fine-tuning LLMs and optimizing non-convex functions pose challenges for Newton's method, which uses the Hessian as a pre-conditioner. The method may converge to local maxima rather than local minima. Moreover, the inaccuracy of Hessian estimates and changes in the Hessian along the optimization trajectory can render second-order information unreliable. To address these issues, we draw inspiration from Sophia. While Sophia performs clipping on the Newton update $\boldsymbol{H}^{-1}\boldsymbol{g}$, we propose a more robust approach by applying **layer-wise clipping directly to the Hessian matrix $\boldsymbol{H}$**.using a universal clipping threshold for the

update $\boldsymbol{H}^{-1}\boldsymbol{g}$ disregards the differences in layer-wise Hessian and gradient magnitudes, which are frequently observed during DNN training, and may distort valuable gradient information. Moreover, $\boldsymbol{H}^{-1}\boldsymbol{g}$ introduces excessive bias, potentially distorting useful gradient information, whereas clipping extreme Hessian values more effectively preserves essential second-order information.

In particular, we improve convergence rates by (1) considering only the positive entries of the diagonal Hessian and (2) introducing per-coordinate, layer-wise clipping of the Hessian values. This approach adapts the clipping threshold across layers to account for the diverse curvature across different parts of the model. Given a clipping threshold $\lambda_i > 0$ for layer $i$, the clipping function is defined as:

$$\text{clip}(\boldsymbol{h}_i) = \max(\boldsymbol{h}_i, \lambda_i), \quad \lambda_i \in \mathbb{R},$$

where all operations are applied element-wise for each layer. The update rule for layer $i$ is then written as:

$$\boldsymbol{\theta}_{t+1,i} = \boldsymbol{\theta}_{t,i} - \eta \cdot \frac{\boldsymbol{m}_{t,i}}{\gamma \cdot \max(\boldsymbol{h}_{t,i}, \lambda_i) + \epsilon},$$

where $\epsilon > 0$ is a small constant to avoid division by zero, and $\lambda_i$ controls the fraction of clipped Hessian values per layer. By applying layer-wise clipping, we ensure that the optimizer is capable of adapting to the curvature of each layer individually, leading to improved stability and convergence rates across different parts of the model. We present the pseudo-code of our Hessian-clipped method in Algorithm 1.

For further information about the differences of HELENE with previous work, please reference the related work in Appendix A.

## 4 CONVERGENCE ANALYSIS

In this section, we provide a theoretical analysis of the convergence of our proposed method. The key improvement in our method comes from the use of layer-wise parameters $\lambda_i$, which reduces the dependency on the total dimension $d$ and instead relies on the maximum layer dimension $\max_i d_i$.

The theoretical bound for the number of steps $T$ in our method is given by the following theorem with two assumptions:

**Assumption 1.** *Let $L : \mathbb{R}^d \to \mathbb{R}$ be a loss function. We assume $L$ is twice continuously differentiable strictly convex, and has a unique minimizer denoted by $\boldsymbol{\theta}^*$. For each layer $i$, define $\mu_i$ as the minimum eigenvalue of the Hessian matrix of $L$ concerning the parameters of that layer evaluated at its minimizer:*

$$\mu_i \equiv \lambda_{\min}(\nabla^2_{\boldsymbol{\theta}_i} L(\boldsymbol{\theta}^*))$$

*where $\nabla^2_{\boldsymbol{\theta}_i}$ denotes the Hessian with respect to the parameters of the $i$-th layer.*

**Assumption 2.** *Regarding the Hessian $\nabla^2 L(\boldsymbol{\theta})$ of the loss function, we assume:*

- *There exists a radius $R_i > 0$ such that for any $\boldsymbol{\theta}_i, \boldsymbol{\theta}'_i \in \mathbb{R}^d$ with $\|\boldsymbol{\theta}_i - \boldsymbol{\theta}'_i\|_2 \leq R_i$, the following inequality holds:*

$$\left\|\nabla^2 L(\boldsymbol{\theta}'_i \mid \boldsymbol{\theta}_{-i})^{-1} \nabla^2 L(\boldsymbol{\theta}_i \mid \boldsymbol{\theta}_{-i})\right\|_2 \leq 2$$

*where $\|\cdot\|_2$ represents the spectral norm.*

**Theorem 1.** *Under Assumptions 1 and 2, let $\eta = \frac{1}{2}$ and $\lambda_i = \frac{R_i}{2\sqrt{d_i}}$. The update reaches a loss at most $\epsilon$ in*

$$T \leq \max_i \left\{ d_i \cdot (L(\boldsymbol{\theta}_{0,i}) - \min L) + \ln\left(\frac{\mu_i R_i^2}{32 d_i \epsilon}\right) \right\}.$$

*steps, where $L$ is the loss function, $\boldsymbol{\theta}_{0,i}$ is the initial parameter vector for layer $i$, $\mu_i$ is the strong convexity constant for layer $i$, and $R_i$ is the bound on the distance between $\boldsymbol{\theta}_{0,i}$ and $\boldsymbol{\theta}^*_i$.*

The best known theoretical bound for the number of steps $T$ required by Sophia to reach a loss at most $\epsilon$ is given by Sophia in which $T_{\text{SOPHIA}} \sim \mathcal{O}(d)$, where $d$ is the total dimension of the parameter space. This result implies that the convergence rate depends linearly on the total dimension $d$, which can lead to slow convergence for models with large parameter spaces. In contrast, our method introduces layer-wise parameters $\rho_i = \frac{R_i}{2\sqrt{d_i}}$, where $R_i$ is the bound on the distance between the initial parameters $\boldsymbol{\theta}_{0,i}$

and the optimal parameters $\boldsymbol{\theta}_i^*$ for layer $i$, and $d_i$ is the dimension of the parameter space for layer $i$. This layer-wise setting significantly reduces the complexity to $T_{\text{SOPHIA}} \sim \mathcal{O}(\max_i d_i)$, which is the maximum dimension across layers. Besides the lower runtime bound, our method allow each layer to have its own parameter $\rho_i$, allowing the method to adapt to the specific geometry of each layer. Refer to Appendix B.3 for the empirical study on the significant variance using unified parameter clipping across different layers. This flexibility leads to a more efficient optimization process, as each layer is treated independently based on its characteristics. In large models where some layers have much smaller dimensions than others, our method is able to achieve faster convergence by focusing on the most difficult layer with the largest dimension, therefore making our method more scalable for deep models with many layers. Detailed proof can be seen in the Appendix C.

## 5 EXPERIMENTS

Since the introduction of the Transformer (Vaswani, 2017), language models (LMs) have progressively developed through the use of different Transformer-based architectures. One of the iconic work is BERT (Devlin, 2018), which is based on the encoder architecture of Transformer and pre-trained with techniques like masked language modeling. As the field of natural language processing (NLP) develops, more powerful decoder-only LLMs also have shown their great potential.

Therefore, to rigorously evaluate the capability and universality of HELENE, we follow the experiments conducted in MeZO on both medium-sized masked LMs (RoBERTa-large (Liu, 2019), 350M) and auto-regressive LLMs (OPT-1.3B (Zhang et al., 2023)) under both few-shot and many-shot settings. Additionally, all optimization algorithms are evaluated with three tuning methods: fine-tuning (FT) and two parameter-efficient fine-tuning (PEFT) methods, LoRA (Hu et al., 2021) and prefix-tuning (Li & Liang, 2021). We also do experiments with zeroth-order (ZO) versions of some optimizers as well as ZO-SGD variants introduced in Zhang et al. (2024), and present them in Section 5.3.

The experimental results show that across all settings, HELENE not only outperforms MeZO on most datasets by approximately 1.5% on average, but also makes the convergence process of gradient-free optimization more stable and faster, boosting to $20\times$ times the original speed.

### 5.1 MASKED LANGUAGE MODELS

For masked LMs, we conduct experiments using RoBERTa-large on three types of NLP tasks, sentiment classification, natural language inference, and topic classification with $k = 16$ examples per class. We run HELENE for 5,000 steps and FT for 1,000 steps. The experimental results are listed in Table 1.

| Task Type | SST-2 | SST-5 | SNLI | MNLI | RTE | TREC |
|---|---|---|---|---|---|---|
| | —— sentiment —— | | —— natural language inference —— | | | — topic — |
| Zero-shot | 79.0 | 35.5 | 50.2 | 48.8 | 51.4 | 32.0 |
| LP | 76.0 ($\pm$2.8) | 40.3 ($\pm$1.9) | 66.0 ($\pm$2.7) | 56.5 ($\pm$2.5) | 59.4 ($\pm$5.3) | 51.3 ($\pm$5.5) |
| FT | 91.9 ($\pm$1.8) | 46.7 ($\pm$1.9) | 77.5 ($\pm$2.6) | 70.0 ($\pm$2.3) | 66.4 ($\pm$7.2) | 85.0 ($\pm$2.5) |
| FT(LoRA) | 91.4 ($\pm$1.7) | 46.7 ($\pm$1.1) | 74.9 ($\pm$4.3) | 67.7 ($\pm$1.4) | 66.1 ($\pm$3.5) | 82.7 ($\pm$4.1) |
| FT(Prefix) | 91.9 ($\pm$1.0) | 47.7 ($\pm$1.1) | 77.2 ($\pm$1.3) | 66.5 ($\pm$2.5) | 66.6 ($\pm$2.0) | 85.7 ($\pm$1.3) |
| MeZO | 90.5 ($\pm$1.2) | 45.5 ($\pm$2.0) | 68.5 ($\pm$3.9) | 58.7 ($\pm$2.5) | 64.0 ($\pm$3.3) | 76.9 ($\pm$2.7) |
| MeZO (LoRA) | 91.4 ($\pm$0.9) | 43.0 ($\pm$1.6) | 69.7 ($\pm$6.0) | 64.0 ($\pm$2.5) | 64.9 ($\pm$3.6) | 73.1 ($\pm$6.5) |
| MeZO (Prefix) | 90.8 ($\pm$1.7) | 45.8 ($\pm$2.0) | 71.6 ($\pm$2.5) | 63.4 ($\pm$1.8) | 65.4 ($\pm$3.9) | 80.3 ($\pm$3.6) |
| HELENE | **92.6** ($\pm$2.3) | **46.7** ($\pm$0.8) | **72.0** ($\pm$2.6) | **58.9** ($\pm$1.1) | **65.7** ($\pm$1.2) | **78.1** ($\pm$1.5) |
| HELENE (LoRA) | 90.6 ($\pm$0.3) | 41.8 ($\pm$1.0) | 68.5 ($\pm$2.0) | 59.0 ($\pm$1.1) | **66.8** ($\pm$3.2) | 67.4 ($\pm$2.1) |
| HELENE (Prefix) | **91.7** ($\pm$0.6) | **46.0** ($\pm$0.7) | 69.5 ($\pm$1.9) | **64.6** ($\pm$2.1) | **66.1** ($\pm$1.8) | 77.4 ($\pm$2.1) |

Table 1: Experiments on RoBERTa-large (350M parameters, $k = 16$). PEFT represents the LoRA and prefix and we report the best of them. All reported numbers are averaged accuracy (standard deviation) across 5 runs.

| Task | SST-2 | RTE | CB | BoolQ | WSC | WIC | COPA | ReCoRD | SQuAD |
|---|---|---|---|---|---|---|---|---|---|
| Task type | | | classification | | | | multiple choice | | – generation – |
| Zero-shot | 53.4 | 53.1 | 37.5 | 45.7 | 44.2 | 57.0 | 75.0 | 70.3 | 27.1 |
| ICL | 80.3 | 53.1 | 48.2 | 58.5 | 44.2 | 50.6 | 69.0 | 71.0 | 59.0 |
| LP | 80.3 | 52.7 | 44.6 | 58.9 | 47.1 | 50.6 | 69.0 | 71 | 75.9 |
| MeZO | 89.6 | 55.8 | 77.0 | 59.6 | 55.0 | 58.0 | 74.0 | 60.0 | 62.2 |
| MeZO (LoRA) | 90.8 | 63.0 | 78.0 | 67.2 | 51.2 | 58.0 | 79.0 | 59.8 | 67.6 |
| MeZO (prefix) | 92.4 | 52.8 | 66.0 | 61.6 | 51.6 | 52.8 | 74.0 | 56.8 | 56.0 |
| HELENE | **91.2** | **64.4** | **87.0** | 60.8 | 55.4 | **58.4** | 69.0 | 55.6 | **63.8** |
| HELENE (LoRA) | **91.4** | 50.6 | 76 | 64.0 | 49.6 | 52.6 | **82.0** | **60.2** | 60.4 |
| HELENE (prefix) | **92.4** | 51.6 | **74.0** | 62.5 | **52** | **57.2** | **80.0** | 58.8 | **68.4** |
| FT (12× memory) | 90.8 | 73.4 | 77 | 70.2 | 53 | 60.2 | 81.0 | 59.6 | 70.9 |

Table 2: Experiments on OPT-1.3B (with 1000 examples). ICL: in-context learning; LP: linear probing; FT: full-parameter fine-tuning with Adam. We highlight the best results in bold to facilitate comparison.

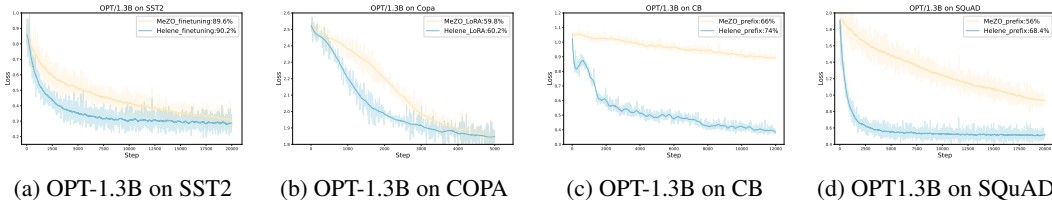

(a) OPT-1.3B on SST2   (b) OPT-1.3B on COPA   (c) OPT-1.3B on CB   (d) OPT1.3B on SQuAD

Figure 3: Performance and convergence of MeZO and HELENE for fine-tuning, LoRA, and prefix-tuning of OPT-1.3B on different datasets. HELENE achieves approximate 10× speedup and up to 15% accuracy improvement compared to MeZO.

**HELENE largely outperforms zero-shot and linear probing.** On all six datasets, HELENE can stably optimize the pre-trained LM and consistently perform better than zero-shot and linear probing.

**HELENE delivers a 20× speed improvement over MeZO while also maintaining its performance.** With the guidance of layer-wise clipped Hessian information, HELENE can reach convergence in about 5000 steps on average across the datasets, accelerating the optimization process by approximate 20× times than MeZO. Meanwhile, the results show that HELENE can still achieve performance on par with MeZO, with leading average accuracy of three tuning methods on the dataset SST-2, SST-5, MNLI and RTE.

## 5.2 AUTO-REGRESSIVE LLMS

LLMs such as GPT-3 (Brown et al., 2020), LLaMA (Touvron et al., 2023), and ChatGLM (Du et al., 2021) have become the predominant models in NLP, we include experiments with auto-regressive LLM OPT-1.3B on three different task: text classification, multiple choice, and text generation. We use various datasets from the SuperGLUE benchmark (Wang et al., 2019), which includes the following datasets: SST-2, RTE, CB, WSC, WIC, COPA, and ReCoRD. Additionally, we also experiment on BoolQ (Clark et al., 2019) and SQuAD (Rajpurkar, 2016). We run HELENE with about 10,000 training steps for each dataset. The results are summarized in Table 2, from which we can have the following observations.

**HELENE has clear performance advantages compared with MeZO.** Table 2 shows that HELENE with its LoRA and prefix variants can consistently outperform MeZO. Specifically, the average performances of HELENE with its LoRA and prefix variants remarkably exceed MeZO's by 5.3%, 2.1% and 1.3% on CB, SQuAD and COPA, respectively.

**HELENE accelerates 10× times while remaining compatible with PEFT methods.** In Figure 3, we present results from four selected datasets across different tasks under three tuning methods. It indicates that HELENE can consistently speed up the convergence by up to 10× times, and also enhances the capability.

| SST2 | Roberta-Large | | | OPT-1.3B | | |
|---|---|---|---|---|---|---|
| | FT | LoRA | Prefix | FT | LoRA | Prefix |
| FO-SGD | 91.4 | 91.2 | 89.6 | 91.1 | 93.6 | 93.1 |
| Forward-Grad | 90.1 | 89.7 | 89.5 | 90.3 | 90.3 | 90.0 |
| ZO-SGD | 89.4 | 90.8 | 90.0 | 90.8 | 90.1 | 91.4 |
| ZO-SGD-MMT | 89.6 | 90.9 | 90.1 | 85.2 | 91.3 | 91.2 |
| ZO-SGD-Cons | 89.6 | **91.6** | 90.1 | 88.3 | 90.5 | 81.8 |
| ZO-SGD-Sign | 52.5 | 90.2 | 53.6 | 87.2 | 91.5 | 89.5 |
| ZO-Adam | 89.8 | 89.5 | 90.2 | 84.4 | **92.3** | 91.4 |
| HELENE | **92.6** | 90.6 | **91.7** | **90.8** | 91.4 | **92.4** |

Table 3: Performance of LLM fine-tuning on SST2 over pre-trained Roberta-Large and OPT-1.3B. Best performance among ZO methods (including Forward-Grad) are in **bold**.

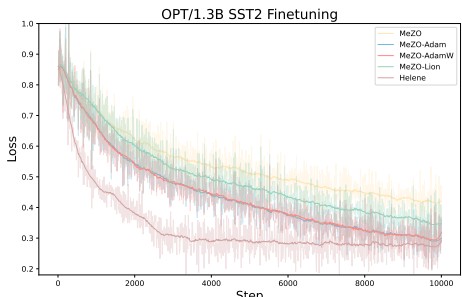

Figure 4: Validation losses for ZO-optimizers. MeZO:0.426, Adam:0.286, AdamW:0.351, Lion:0.343, HELENE:0.283.

### 5.3 EXPERIMENTS WITH OTHER ZO ALGORITHMS

It is worth noting that the ZO optimization technique utilized in Malladi et al. (2023) is primarily the basic SGD version (ZO-SGD), and it is still not clear how effective HELENE is when comparing with other ZO optimization algorithms like ZO-SGD, ZO-SGD-MMT, ZO-SGD-Cons, ZO-SGD-Sign and ZO-Adam as introduced in Liu et al. (2020). Therefore, we reference the statistics of performances summarized in Zhang et al. (2024) and experiment under the same setting with them (Table 3), through which HELENE shows good functionality especially for FT and prefix-tuning.

We further implement the ZO versions of Adam, AdamW and Lion (Chen et al., 2024) and plot the results in Figure 4. The results indicate that HELENE helps the model converge faster as well as obtain lower validation loss value.

### 5.4 ABLATION STUDY

We conduct a comprehensive ablation study on the key techniques of HELENE in Appendix B, including in-depth analysis of the effects of magnitude clipping across different ranges. Additionally, we explore the factors resulting in Sophia's initial convergence and subsequent divergence.

## 6 CONCLUSION

In this paper, we present a novel optimizer, HELENE, which is designed to address the challenges of fine-tuning LLMs. HELENE integrates a new asymptotic Gauss-Newton-Bartlett (A-GNB) estimator for diagonal Hessian estimation, and a novel layer-wise clipping with the annealing module. The A-GNB estimator eliminates the need for label sampling, providing an unbiased Hessian approximation and improving the precision of curvature-aware updates. Furthermore, our layer-wise clipping mechanism provably ensures more adaptive Hessian updates based on the curvature of each layer, enhancing stability and scalability. Theoretical analysis shows that HELENE reduces convergence steps from $\mathcal{O}(d)$ to $\mathcal{O}(\max_i d_i)$, making it highly scalable for modern architectures with many layers. Experimental results on models like RoBERTa-large and OPT-1.3B demonstrate that HELENE achieves up to a $20\times$ speedup compared to MeZO and improves performance by 1.5% on average. Compatible with both full parameter tuning and parameter-efficient fine-tuning, HELENE outperforms many state-of-the-art optimizers across diverse tasks and datasets.

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

# A    RELATED WORK

## A.1    ZERO-ORDER OPTIMIZATION

Zeroth-order (ZO) optimization, which only relies on the forward passes of neural networks, offers significant memory savings during the training process. Recently, MeZO (Malladi et al., 2023) adapted the traditional zeroth-order SGD optimization method for fine-tuning LMs, achieving performance comparable to full-parameter fine-tuning while significantly reducing memory usage. Thus, zeroth-order optimization is regarded as a promising approach for memory-efficient fine-tuning of LLMs. Several studies have aimed to improve the MeZO algorithm. For instance, Gautam et al. (2024) introduced a zeroth-order optimization algorithm that integrates both full-batch and mini-batch information to produce asymptotically unbiased, low-variance gradient estimations. However, the convergence rate of their approach still leaves room for improvement. In pursuit of better gradient estimation, Jiang et al. (2024) proposed an innovative perturbation sampling technique inspired by the Adam optimizer. Other methods, such as SPSA (Spall, 1992; Maryak & Chin, 2001), have shown to be effective in non-convex multi-agent optimization (Tang et al., 2020; Hajinezhad & Zavlanos, 2018) and in generating black-box adversarial examples (Chen et al., 2017; Cai et al., 2021; Liu et al., 2019; Ye et al., 2018).

## A.2    SECOND-ORDER INFORMATION FOR FINE-TUNING LLMS

Classic second-order optimization algorithms pre-condition the gradient with curvature information (BROYDEN, 1970; Nesterov & Polyak, 2006; Conn et al., 2000). Over the years, people have developed numerous ways to adapt these methods to deep learning. To the best of our knowledge, BECKER (1988) was the first to use diagonal Hessian as the pre-conditioner. Martens et al. (2010) approximated the Hessian with conjugate gradient. Schaul et al. (2013) automatically tuned the learning rate of SGD by considering diagonal Hessian. Pascanu (2013) considered Gaussian Newton's approximation of Hessian and Fisher information matrix. Martens & Grosse (2015) and follow-up works (Ba et al., 2017; George et al., 2018; Martens et al., 2018; Zhang et al., 2022) proposed to approximate the Hessian based on the structure of neural networks. Despite these progress on deep learning applications, for decoder-only LLMs, Adam still appears to be the most popular optimizer. The authors of this paper suspect that many previous second-order optimizers face the challenge that the computational / memory overhead due to frequent Hessian computation hinders improvements in wall-clock time (Martens & Grosse, 2015; Gupta et al., 2018). Some of them also depend on specific model architecture or hardware structures, e.g., Anil et al. (2020) offloads hessian computation to CPUs, and George et al. (2018) needs ResNet and very large batch size to approximate the Fisher information matrix. To the best of our knowledge, there was no previous report that second-order optimizers can achieve a speed-up on LLMs in total compute.

There is also a concurrent work HiZOO (Zhao et al., 2024) that utilizes Hessian information to enhance zeroth-order optimization for fine-tuning LLMs. A major focus of HiZOO is to introduce one more forward pass to handle heterogeneous curvatures across parameter dimensions. However, our work focus on incorporating layer-wise clipping to exclude extreme Hessian values and Exponential Moving Average (EMA) to improve generalization.

## A.3    GRADIENT CLIPPING

Global gradient clipping has been a widely adopted practice in fine-tuning LLMs (Chen et al., 2020; Zhang et al., 2019; Liu et al., 2022). This technique stabilizes training by mitigating the impact of rare examples and large gradient noise. In addition to gradient clipping, HELENE is the first method to clip the Hessian matrix in second-order optimization techniques. This approach addresses the issue of the Hessian matrix fluctuating along the optimization trajectory and reduces the errors in Hessian approximations.

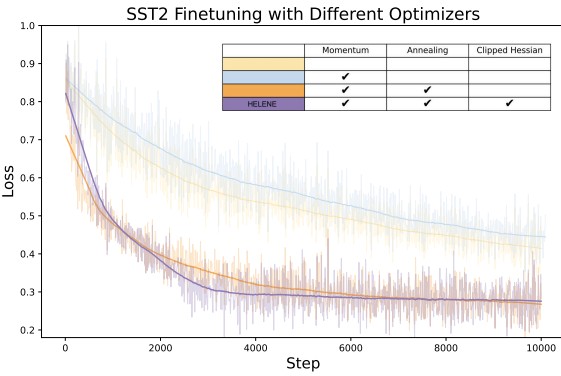 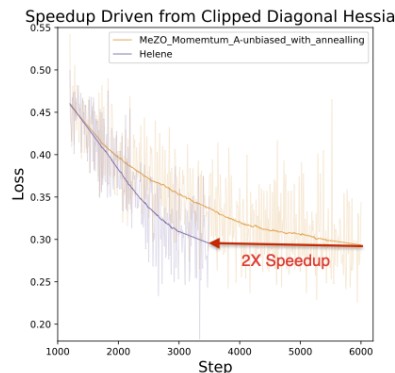

(a) Ablation study for key components of HELENE. The effectiveness of annealing and the clipped diagonal Hessian are demonstrated progressively from high to low.

(b) The image on the right is a zoomed-in view of the left one, showing that our proposed HELENE is twice as fast as the compared methods.

Figure 5: Comparison of tuning processes and ablation studies with different optimization algorithms.

# B ABLATION STUDY

## B.1 EVALUATING THE IMPACT OF KEY COMPONENTS ON CONVERGENCE AND STABILITY

Figure 5 illustrates the effectiveness of each component in our algorithm. Adding momentum to MeZO alone doesn't improve performance. Introducing bias in the gradient boosts convergence speed, but causes loss to increase later in training due to biased gradient estimates. To counter this, we added an annealing term to make the gradient asymptotically unbiased, which stabilizes the loss. Inspired by Sophia, we introduced the clipped Hessian to address heterogeneous curvatures, further improving convergence speed. Our ablation study validates both the motivation and effectiveness of these components.

## B.2 MAGNITUDE CLIPPING

Figure 6 addresses the robustness of clipping in our optimizer. Our empirical study is as follows: First, we explored the impact of lower bounds ranging from 1 to 3, all of which demonstrated stability. As a hyperparameter, this lower bound shows consistent robustness. However, when the lower bound was set to 0.9, the model performance dropped by 10 points, leading us to believe that problematic Hessian values are concentrated below 1, while values above 1 are less critical. Second, we argue that layer-wise clipping based on magnitude is reasonable in a zeroth-order setting, as performing percentage-based clipping for each layer would be time-consuming.

## B.3 INVESTIGATION INTO THE CONVERGENCE INSTABILITY OF SOPHIA

We study the reasons for Sophia's failure in the Figure 1 by counting the number of clip triggers. We computed the loss between timesteps 400 and 800, with a mean value of 0.57. The average loss between timesteps 1400 and 1800 was 0.65. We then analyzed the number of times the Sophia clipping mechanism was triggered within these two time intervals. Our analysis covered the Q, K, V matrices, fully connected layers, and bias layers. We found that the frequency of clipping in the interval where the mean loss was 0.65 was 1.18 to 1.22 times higher than in the interval where the mean loss was 0.57.

Based on these experimental observations, we conclude that Sophia's clipping mechanism tends to be over-triggered in complex data scenarios, particularly when faced with heterogeneous curvature. This over-triggering can result in non-convergence, aligning with our intuition. In the zeroth-order setting, gradients are estimated using SPSA, and excessive clipping of the $\frac{g}{H}$ terms can lead to instability and failure of the model to converge.

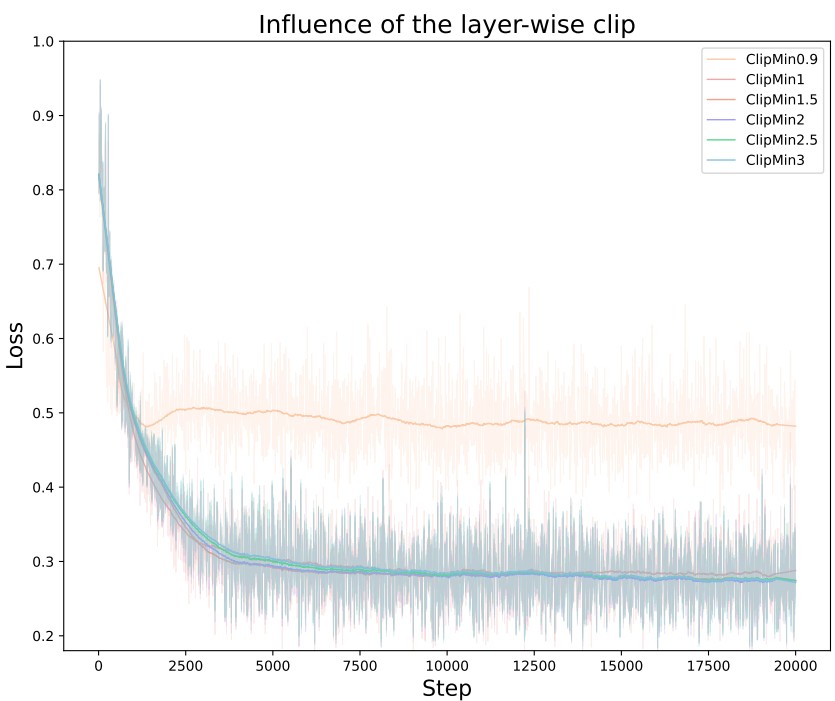

Figure 6: Performance and convergence of HELENE for fine-tuning of OPT-1.3B on SST2 with different clipping lower bound.

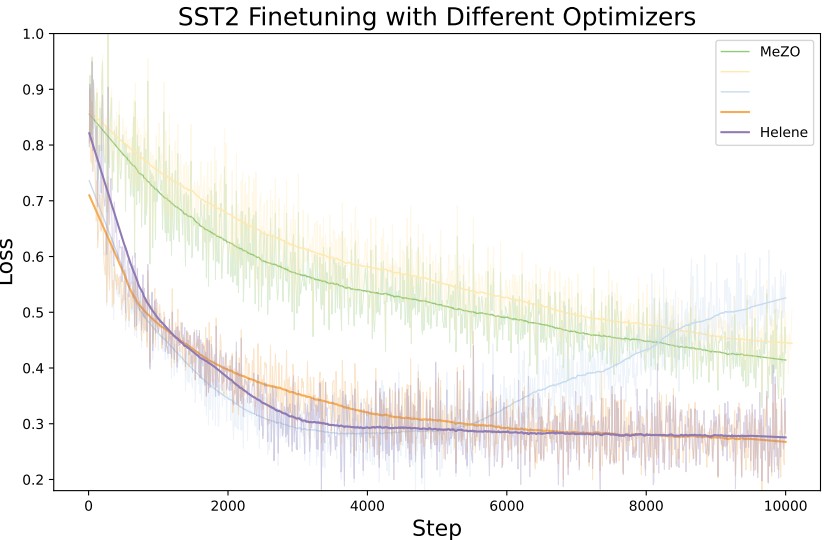

Figure 7: Gradient Descent methods, SPSA gradient estimation becomes increasingly noisy as optimization progresses.

## C DETAILED CONVERGENCE ANALYSIS

**Lemma 1** (Divergence to Infinity). *Under Assumption 1, for each layer $i$ in a neural network model, assume the function $L : \mathbb{R}^{d_i} \to \mathbb{R}$ is strictly convex, twice continuously differentiable, and has a unique minimizer denoted by $\boldsymbol{\theta}_i^*$. For any parameter vector $\boldsymbol{\theta}_i$ of layer $i$ such that $\|\boldsymbol{\theta}_i - \boldsymbol{\theta}_i^*\|_2 \geq 1$, the function $L(\boldsymbol{\theta}_i)$ diverges to infinity as $\|\boldsymbol{\theta}_i\|_2 \to \infty$.*

*Proof.* By the strict convexity of $L$, for any $\boldsymbol{\theta}_i$ such that $\|\boldsymbol{\theta}_i - \boldsymbol{\theta}_i^*\|_2 \geq 1$, we have:

$$\frac{L(\boldsymbol{\theta}_i) - L(\boldsymbol{\theta}_i^*)}{\|\boldsymbol{\theta}_i - \boldsymbol{\theta}_i^*\|_2} \geq \min_{\|\phi\|_2 = 1} L(\boldsymbol{\theta}_i^* + \phi) - L(\boldsymbol{\theta}_i^*), \tag{4}$$

where $\phi$ is a unit vector. For the convenience, here $L(\boldsymbol{\theta}_i)$ denotes $L(\boldsymbol{\theta}_i | \boldsymbol{\theta}_{-i})$ where $\boldsymbol{\theta}_{-i}$ denotes the parameters in the whole model except $\boldsymbol{\theta}_i$, and $L(\boldsymbol{\theta}_i^*)$ denotes $L(\boldsymbol{\theta}_i^* | \boldsymbol{\theta}_{-i}^*)$. Define $\Delta_i$ as:

$$\Delta_i = \min_{\|\phi\|_2 = 1} L(\boldsymbol{\theta}_i^* + \phi) - L(\boldsymbol{\theta}_i^*), \tag{5}$$

a positive constant due to the strict convexity of $L$ indicating the minimal rate of increase of $L$ around $\boldsymbol{\theta}_i^*$.

Thus, the inequality can be written as:

$$L(\boldsymbol{\theta}_i) \geq \|\boldsymbol{\theta}_i - \boldsymbol{\theta}_i^*\|_2 \Delta_i + L(\boldsymbol{\theta}_i^*). \tag{6}$$

This implies that as $\|\boldsymbol{\theta}_i\|_2 \to \infty$, which necessarily implies $\|\boldsymbol{\theta}_i - \boldsymbol{\theta}_i^*\|_2 \to \infty$, the loss $L(\boldsymbol{\theta}_i)$ also diverges to infinity, since the term $\|\boldsymbol{\theta}_i - \boldsymbol{\theta}_i^*\|_2 \Delta_i$ dominates and increases without bound. $\qquad \square$

**Note:** We do not require the Hessian of the loss function, $\nabla^2 L(\boldsymbol{\theta}_i)$, to be Lipschitz continuous; Assumption 2 only requires that the Hessian is continuous in a multiplicative sense within a neighborhood of constant radius.

**Lemma 2** (Parameter Bound). *Let $L : \mathbb{R}^d \to \mathbb{R}$ be a loss function for a neural network composed of multiple layers, each with parameters $\boldsymbol{\theta}_i$, and $L$ is twice continuously differentiable and strictly convex with respect to each layer's parameters at a global minimizer $\boldsymbol{\theta}^*$. Assume each layer $i$ satisfies the following condition:*

$$L(\boldsymbol{\theta}_i) - \min L \leq \frac{\mu_i R_i^2}{4},$$

*where $\mu_i$ is the minimum eigenvalue of the Hessian of $L$ at the minimizer for parameters of layer $i$, and $R_i$ is a predefined radius. Then, it holds that*

$$\|\boldsymbol{\theta}_i - \boldsymbol{\theta}_i^*\|_2 \leq R_i.$$

*Proof.* Suppose, by way of contradiction, there exists a $\boldsymbol{\theta}_i$ such that $L(\boldsymbol{\theta}_i) \leq \frac{\mu_i R_i^2}{4}$, but $\|\boldsymbol{\theta}_i - \boldsymbol{\theta}_i^*\|_2 > R_i$. Define $\boldsymbol{\theta}_i'$ as:

$$\boldsymbol{\theta}_i' = \boldsymbol{\theta}_i^* + \sqrt{2(L(\boldsymbol{\theta}_i) - \min L)} \frac{\boldsymbol{\theta}_i - \boldsymbol{\theta}_i^*}{\mu_i \|\boldsymbol{\theta}_i - \boldsymbol{\theta}_i^*\|_2}. \tag{7}$$

Since $\boldsymbol{\theta}_i'$ is a point between $\boldsymbol{\theta}_i$ and $\boldsymbol{\theta}_i^*$, and due to the strict convexity of $L$, we have $L(\boldsymbol{\theta}_i') < L(\boldsymbol{\theta}_i)$ by convexity. Considering the Taylor expansion of $L$ at $\boldsymbol{\theta}_i^*$ along the direction towards $\boldsymbol{\theta}_i'$, we have:

$$f(t) = L(\boldsymbol{\theta}_i^* + t(\boldsymbol{\theta}_i' - \boldsymbol{\theta}_i^*)), \quad f(1) = L(\boldsymbol{\theta}_i'), \quad f(0) = L(\boldsymbol{\theta}_i^*), \quad f'(0) = 0, \tag{8}$$

$$f''(t) = (\boldsymbol{\theta}_i' - \boldsymbol{\theta}_i^*)^T \nabla^2 L(t\boldsymbol{\theta}_i' + (1-t)\boldsymbol{\theta}_i^*)(\boldsymbol{\theta}_i' - \boldsymbol{\theta}_i^*), \tag{9}$$

Given $f''(t) \geq \frac{\mu_i}{2} \|\boldsymbol{\theta}_i' - \boldsymbol{\theta}_i^*\|_2^2$ from Assumption 2 and the convexity, the Taylor expansion yields:

$$f(1) \geq f(0) + f'(0) + \frac{1}{2} f''(t) = L(\boldsymbol{\theta}_i^*) + \frac{\mu_i}{2} \|\boldsymbol{\theta}_i' - \boldsymbol{\theta}_i^*\|_2^2,$$

thus,

$$L(\boldsymbol{\theta}_i') \geq L(\boldsymbol{\theta}_i^*) + \frac{\mu_i}{2} \|\boldsymbol{\theta}_i' - \boldsymbol{\theta}_i^*\|_2^2$$

which contradicts the assumption that $L(\boldsymbol{\theta}_i') < L(\boldsymbol{\theta}_i)$. Therefore, the original assumption that $\|\boldsymbol{\theta}_i - \boldsymbol{\theta}_i^*\|_2 > R_i$ must be false, concluding that $\|\boldsymbol{\theta}_i - \boldsymbol{\theta}_i^*\|_2 \leq R_i$. $\qquad \square$

**Lemma 3** ( Gradient Norm Bound). *For any $\boldsymbol{\theta}_i$ in layer $i$ of a neural network, satisfying $\|\nabla L(\boldsymbol{\theta}_i)\|_2 \leq \frac{\mu_i R_i}{2}$, where $\mu_i$ is the minimum eigenvalue of the Hessian of $L$ at the minimizer for layer $i$ parameters, it holds that $\|\boldsymbol{\theta}_i - \boldsymbol{\theta}_i^*\|_2 \leq R_i$.*

*Proof.* Assume, by way of contradiction, there exists a $\boldsymbol{\theta}_i$ with $\|\nabla L(\boldsymbol{\theta}_i)\|_2 \leq \frac{\mu_i R_i}{2}$ and $\|\boldsymbol{\theta}_i - \boldsymbol{\theta}_i^*\|_2 > R_i$. Define a function $f(t)$ by:

$$f(t) = \nabla L(\boldsymbol{\theta}_i^* + t \cdot (\boldsymbol{\theta}_i - \boldsymbol{\theta}_i^*)) \cdot \frac{\boldsymbol{\theta}_i - \boldsymbol{\theta}_i^*}{\|\boldsymbol{\theta}_i - \boldsymbol{\theta}_i^*\|_2}, \tag{10}$$

where $f(0) = \nabla L(\boldsymbol{\theta}_i^*)$ due to $\boldsymbol{\theta}_i^*$ being a minimizer, and $f(R_i) = \nabla L(\boldsymbol{\theta}_i)$.

Due to the strict convexity of $L$, $f(t)$ is a strictly monotone increasing function. The derivative with respect to $t$, must satisfy:

$$f'(t) = \frac{d}{dt}\left(\nabla L(\boldsymbol{\theta}_i^* + t \cdot (\boldsymbol{\theta}_i - \boldsymbol{\theta}_i^*)) \cdot \frac{\boldsymbol{\theta}_i - \boldsymbol{\theta}_i^*}{\|\boldsymbol{\theta}_i - \boldsymbol{\theta}_i^*\|_2}\right) \geq \frac{\|\nabla L(\boldsymbol{\theta}_i)\|_2}{2}, \tag{11}$$

by Assumption 2 and the fact that the gradient norm does not increase more than twice in any direction within the ball of radius $R_i$.

The fundamental theorem of calculus and the above inequality imply:

$$f(R_i) = f(0) + \int_0^{R_i} f'(t)dt \geq \int_0^{R_i} \frac{\|\nabla L(\boldsymbol{\theta}_i)\|_2}{2}dt = \frac{R_i\|\nabla L(\boldsymbol{\theta}_i)\|_2}{2}, \tag{12}$$

However, $f(R_i) = \|\nabla L(\boldsymbol{\theta}_i)\|_2$ and this leads to

$$\|\nabla L(\boldsymbol{\theta}_i)\|_2 \geq \frac{R_i\|\nabla L(\boldsymbol{\theta}_i)\|_2}{2}, \tag{13}$$

a contradiction unless $\|\boldsymbol{\theta}_i - \boldsymbol{\theta}_i^*\|_2 \leq R_i$.

Therefore, the original assumption that $\|\boldsymbol{\theta}_i - \boldsymbol{\theta}_i^*\|_2 > R_i$ must be false, proving the lemma. $\qquad\square$

**Lemma 4** (Stability of Gradient Flow). *Suppose the gradient $\nabla L(\boldsymbol{\theta}_i(t))$ and the Hessian $\nabla^2 L(\boldsymbol{\theta}_i(t))$ of the loss function $L$ satisfy the conditions for all $t \in [0, 1]$ that ensure stability and convergence to a minimizer $\boldsymbol{\theta}_i^*$. Assume the differential equation*

$$\frac{d\boldsymbol{\theta}_i(t)}{dt} = -\left(\nabla^2 L(\boldsymbol{\theta}_i(t))\right)^{-1} \nabla L(\boldsymbol{\theta}_i(t)), \quad \boldsymbol{\theta}_i(0) = \boldsymbol{\theta}_i, \quad \boldsymbol{\theta}_i(1) = \boldsymbol{\theta}_i^*, \tag{14}$$

*has at least one solution on the interval [0, 1] and satisfies $\nabla L(\boldsymbol{\theta}_i(t)) = (1 - t)\nabla L(\boldsymbol{\theta}_i)$ for all $t \in [0, 1]$.*

*Proof.* We demonstrate this by showing that the given ordinary differential equation (ODE) is well-posed under the assumptions. The initial value problem

$$\frac{d\boldsymbol{\theta}_i(t)}{dt} = -\left(\nabla^2 L(\boldsymbol{\theta}_i(t))\right)^{-1} \nabla L(\boldsymbol{\theta}_i(t)), \tag{15}$$

can be solved over the interval [0, 1] due to the continuity and positive definiteness of $\nabla^2 L$, which ensures the existence and uniqueness of the solution by the Picard-Lindelöf theorem.

Define $T_{\max,i}$ as the largest positive number such that the solution exists on $[0, T_{\max,i}]$. We claim $T_{\max,i} \geq 1$, based on the behavior of the gradient along the solution path. Considering:

$$\frac{d}{dt}\nabla L(\boldsymbol{\theta}_i(t)) = \nabla^2 L(\boldsymbol{\theta}_i(t))\frac{d\boldsymbol{\theta}_i(t)}{dt} = -\nabla L(\boldsymbol{\theta}_i(t)), \tag{16}$$

which implies that $\nabla L(\boldsymbol{\theta}_i(t)) = e^{-t}\nabla L(\boldsymbol{\theta}_i)$. Since $\nabla L(\boldsymbol{\theta}_i(t)) = (1 - t)\nabla L(\boldsymbol{\theta}_i)$ for $t \in [0, 1]$, the condition aligns perfectly.

Finally, since $\boldsymbol{\theta}_i(1)$ has zero gradient by the construction of the ODE, $\boldsymbol{\theta}_i(1)$ must be $\boldsymbol{\theta}_i^*$. This completes the proof. $\qquad\square$

**Lemma 5** (Quadratic Form Integration). *Assume the gradient norm $\|\nabla L(\boldsymbol{\theta}_i)\|_2$ and the Hessian $\nabla^2 L(\boldsymbol{\theta}_i)$ satisfy certain conditions over the interval $[0, 1]$. Suppose either*

    *1. $L(\boldsymbol{\theta}_i) - \min L \leq \frac{\mu_i R_i^2}{16}$, or*

2. $\|\nabla L(\boldsymbol{\theta}_i)\|_2 \le \frac{\mu_i R_i}{4}$,

*where $\mu_i$ is the minimum eigenvalue of the Hessian at the minimizer for the parameters of layer $i$, then it holds that*

$$\left\|\nabla L(\boldsymbol{\theta}_i)^T (\nabla^2 L(\boldsymbol{\theta}_i))^{-1} \nabla L(\boldsymbol{\theta}_i)\right\| \le 4(L(\boldsymbol{\theta}_i) - \min L). \tag{17}$$

*Proof.* Let $\{\boldsymbol{\theta}_i(t)\}_{t=0}^1$ be the solution of the following differential equation:

$$\frac{d\boldsymbol{\theta}_i(t)}{dt} = -(\nabla^2 L(\boldsymbol{\theta}_i(t)))^{-1} \nabla L(\boldsymbol{\theta}_i(t)), \quad \boldsymbol{\theta}_i(0) = \boldsymbol{\theta}_i, \quad \boldsymbol{\theta}_i(1) = \boldsymbol{\theta}_i^*. \tag{18}$$

From Lemma 4, adapted for each layer, we have $\nabla L(\boldsymbol{\theta}_i(t)) = (1-t)\nabla L(\boldsymbol{\theta}_i)$ for all $t \in [0,1]$. Assume $\|\boldsymbol{\theta}_i(t) - \boldsymbol{\theta}_i^*\| \le R_i/2$ by Lemmas 2 and 3.

By Assumption 2, for each layer, this implies:

$$(\nabla^2 L(\boldsymbol{\theta}_i(t)))^{-1} \ge \frac{1}{2}(\nabla^2 L(\boldsymbol{\theta}_i))^{-1} \tag{19}$$

for all $t \in [0,1]$.

Integrating the quadratic form along the path, we have:

$$L(\boldsymbol{\theta}_i) - \min L = L(\boldsymbol{\theta}_i(0)) - L(\boldsymbol{\theta}_i(1))$$

$$= \int_0^1 (1-t)^2 (\nabla L(\boldsymbol{\theta}_i))^T (\nabla^2 L(\boldsymbol{\theta}_i(t)))^{-1} \nabla L(\boldsymbol{\theta}_i) dt. \tag{20}$$

Substituting the inequality from equation 19, we simplify:

$$\frac{1}{2} \int_0^1 (1-t)^2 dt (\nabla L(\boldsymbol{\theta}_i))^T (\nabla^2 L(\boldsymbol{\theta}_i))^{-1} \nabla L(\boldsymbol{\theta}_i)$$
$$= \frac{1}{6}(\nabla L(\boldsymbol{\theta}_i))^T (\nabla^2 L(\boldsymbol{\theta}_i))^{-1} \nabla L(\boldsymbol{\theta}_i). \tag{21}$$

This integration shows that $\left\|\nabla L(\boldsymbol{\theta}_i)^T (\nabla^2 L(\boldsymbol{\theta}_i))^{-1} \nabla L(\boldsymbol{\theta}_i)\right\| \le 4(L(\boldsymbol{\theta}_i) - \min L)$, completing the proof. $\qquad\square$

**Lemma 6** (Gradient and Loss Bound). *Assuming the gradient norm $\|\nabla L(\boldsymbol{\theta}_i)\|_2$ and the conditions on the loss function $L$ are such that either*

1. $L(\boldsymbol{\theta}_i) - \min L \le \frac{\mu_i R_i^2}{4}$, *or*

2. $\|\nabla L(\boldsymbol{\theta}_i)\|_2 \le \frac{R_i \mu_i}{2}$,

*it holds that*

$$L(\boldsymbol{\theta}_i) - \min L \le \frac{1}{\mu_i} \|\nabla L(\boldsymbol{\theta}_i)\|^2. \tag{22}$$

*Proof.* The proof follows a reasoning similar to that of Lemma 5 but adapted for each layer. Given the conditions on $L(\boldsymbol{\theta}_i) - \min L$ or the norm of the gradient $\|\nabla L(\boldsymbol{\theta}_i)\|_2$, we utilize the connection between the gradient norm and the difference in loss to bound $L(\boldsymbol{\theta}_i) - \min L$.

From the gradient norm bound $\|\nabla L(\boldsymbol{\theta}_i)\|_2$ and the positive definiteness and continuity of $\nabla^2 L$, the loss function exhibits quadratic behavior near the minimizer. This is characterized by the Taylor expansion:

$$L(\boldsymbol{\theta}_i) \approx L(\boldsymbol{\theta}_i^*) + \frac{1}{2}(\boldsymbol{\theta}_i - \boldsymbol{\theta}_i^*)^T \nabla^2 L(\boldsymbol{\theta}_i^*)(\boldsymbol{\theta}_i - \boldsymbol{\theta}_i^*), \tag{23}$$

where $\boldsymbol{\theta}_i^*$ is the minimizer of $L$.

Using the bound $\|\nabla L(\boldsymbol{\theta}_i)\|_2 \le \frac{R_i \mu_i}{2}$, the Taylor series expansion around $\boldsymbol{\theta}_i^*$ implies:

$$L(\boldsymbol{\theta}_i) - \min L \le \frac{1}{2\mu_i} \|\nabla L(\boldsymbol{\theta}_i)\|^2, \tag{24}$$

satisfying the condition given by Lemma 6.

This completes the proof by relating the behavior of the loss function's gradient at $\boldsymbol{\theta}_i$ to its minimum value, leveraging the quadratic approximation provided by the Hessian at the minimizer. $\qquad\square$

**Lemma 7** (Norm Bound on Inverse Hessian and Gradient Product). *Assuming the gradient $\nabla L(\boldsymbol{\theta}_i)$ and the Hessian $\nabla^2 L(\boldsymbol{\theta}_i)$ satisfy certain conditions such that either*

1. $L(\boldsymbol{\theta}_i) - \min L \leq \frac{\mu_i R_i^2}{16}$, *or*

2. $\|\nabla L(\boldsymbol{\theta}_i)\|_2 \leq \frac{R_i \mu_i}{4}$,

*it holds that*

$$\|\nabla^2 L(\boldsymbol{\theta}_i)^{-1} \nabla L(\boldsymbol{\theta}_i)\|_2 \leq \frac{8(L(\boldsymbol{\theta}_i) - \min L)}{\mu_i}. \tag{25}$$

*Proof.* We derive this by using the properties of the Hessian and the gradient for the loss function $L$ specific to layer $i$. From Lemma 2, we have:

$$\|\boldsymbol{\theta}_i - \boldsymbol{\theta}_i^*\|_2 \leq R_i.$$

Given that $\nabla^2 L(\boldsymbol{\theta}_i^*) \geq \frac{\mu_i}{2} I$, and from Lemma 5 adapted for layer $i$, it holds that:

$$4(L(\boldsymbol{\theta}_i) - \min L) \geq \|\nabla L(\boldsymbol{\theta}_i)^T (\nabla^2 L(\boldsymbol{\theta}_i))^{-1} \nabla L(\boldsymbol{\theta}_i)\|.$$

Expanding and manipulating the inequality, we derive:

$$\|\nabla L(\boldsymbol{\theta}_i)\|^T (\nabla^2 L(\boldsymbol{\theta}_i))^{-1} \|\nabla L(\boldsymbol{\theta}_i)\| = \|\nabla^2 L(\boldsymbol{\theta}_i)^{-1} \nabla L(\boldsymbol{\theta}_i)\|^2.$$

Given $\nabla^2 L(\boldsymbol{\theta}_i)^{-1} \leq \frac{2}{\mu_i} I$, we can substitute this into our calculation to find:

$$\|\nabla^2 L(\boldsymbol{\theta}_i)^{-1} \nabla L(\boldsymbol{\theta}_i)\|^2 \leq \frac{2}{\mu_i} \|\nabla L(\boldsymbol{\theta}_i)\|^2 \leq \frac{4(L(\boldsymbol{\theta}_i) - \min L)}{\mu_i},$$

and finally,

$$\|\nabla^2 L(\boldsymbol{\theta}_i)^{-1} \nabla L(\boldsymbol{\theta}_i)\|_2 \leq \frac{8(L(\boldsymbol{\theta}_i) - \min L)}{\mu_i},$$

completing the proof. $\qquad\square$

**Lemma 8.** *For any $\boldsymbol{\theta}_i \in \mathbb{R}^{d_i}$, where $\boldsymbol{\theta}_i$ represents the parameters for the $i$-th layer, and satisfying that*

$$\|\nabla^2 L(\boldsymbol{\theta}_i)^{-1} \nabla L(\boldsymbol{\theta}_i)\|_2 \leq \frac{R}{2},$$

*it holds that*

$$L(\boldsymbol{\theta}_i) - \min L \leq \nabla L(\boldsymbol{\theta}_i)^T (\nabla^2 L(\boldsymbol{\theta}_i))^{-1} \nabla L(\boldsymbol{\theta}_i) \leq 4(L(\boldsymbol{\theta}_i) - \min L).$$

*Proof.* Let $\{\boldsymbol{\theta}_i(t)\}_{t=0}^1$ be the solution of the following differential equation:

$$\frac{d\boldsymbol{\theta}_i(t)}{dt} = -(\nabla^2 L(\boldsymbol{\theta}_i(t)))^{-1} \nabla L(\boldsymbol{\theta}_i(t)),$$

where $\boldsymbol{\theta}_i(0) = \boldsymbol{\theta}_i$ and $\boldsymbol{\theta}_i(1) = \boldsymbol{\theta}_i^*$.

We claim that for all $t \in [0, 1]$, $\|\boldsymbol{\theta}_i(t) - \boldsymbol{\theta}_i\|_2 \leq R_i$. If not, let $T$ be the smallest positive number such that $\|\boldsymbol{\theta}_i(T) - \boldsymbol{\theta}_i\|_2 = R_i$. Such $T$ exists because $\|\boldsymbol{\theta}_i(t) - \boldsymbol{\theta}_i\|_2$ is continuous in $t$ and $\|\boldsymbol{\theta}_i(0) - \boldsymbol{\theta}_i\|_2 = 0$.

We can now bound the distance:

$$R_i = \|\boldsymbol{\theta}_i(T) - \boldsymbol{\theta}_i(0)\|_2 \leq \int_0^T \left\| \frac{d\boldsymbol{\theta}_i(t)}{dt} \right\|_2 dt.$$

Substituting the derivative expression for $\boldsymbol{\theta}_i(t)$, we get:

$$= \int_0^T \left\| (\nabla^2 L(\boldsymbol{\theta}_i(t)))^{-1} \nabla L(\boldsymbol{\theta}_i(t)) \right\|_2 dt$$

$$\leq \int_0^T \|\nabla^2 L(\boldsymbol{\theta}_i(t))^{-1}\|_2 \|\nabla L(\boldsymbol{\theta}_i(t))\|_2 dt.$$

From Assumption 2, we know that:

$$\nabla^2 L(\boldsymbol{\theta}_i)^{-1} \leq 2(\nabla^2 L(\boldsymbol{\theta}_i(t)))^{-1}.$$

Thus, we can bound this integral:

$$\leq 2 \int_0^T \|\nabla^2 L(\boldsymbol{\theta}_i(t))^{-1} \nabla L(\boldsymbol{\theta}_i(t))\|_2 dt$$

$$\leq 2T \|\nabla^2 L(\boldsymbol{\theta}_i(t))^{-1} \nabla L(\boldsymbol{\theta}_i(t))\|_2.$$

Using the assumption that $\|\nabla^2 L(\boldsymbol{\theta}_i)^{-1} \nabla L(\boldsymbol{\theta}_i)\|_2 \leq \frac{R_i}{2}$, we get:

$$\leq 2T \frac{R_i}{2} = R_i T,$$

which implies that $T = 1$.

Therefore, for all $t \in [0,1]$, $\|\boldsymbol{\theta}_i(t) - \boldsymbol{\theta}_i\|_2 \leq R_i$. By Assumption 2, we also have:

$$2(\nabla^2 L(\boldsymbol{\theta}_i))^{-1} \preceq (\nabla^2 L(\boldsymbol{\theta}_i(t)))^{-1} \preceq \frac{1}{2}(\nabla^2 L(\boldsymbol{\theta}_i))^{-1}.$$

Now, we compute the difference in the loss function:

$$L(\boldsymbol{\theta}_i) - \min L = L(\boldsymbol{\theta}_i(0)) - L(\boldsymbol{\theta}_i(1)) = \int_0^1 \nabla L(\boldsymbol{\theta}_i(t))^T (\nabla^2 L(\boldsymbol{\theta}_i(t)))^{-1} \nabla L(\boldsymbol{\theta}_i(t)) dt.$$

$$= \int_0^1 (1-t) \nabla L(\boldsymbol{\theta}_i)^T (\nabla^2 L(\boldsymbol{\theta}_i))^{-1} \nabla L(\boldsymbol{\theta}_i) dt.$$

Thus:

$$L(\boldsymbol{\theta}_i) - \min L \leq \frac{1}{2} \nabla L(\boldsymbol{\theta}_i)^T (\nabla^2 L(\boldsymbol{\theta}_i))^{-1} \nabla L(\boldsymbol{\theta}_i).$$

Finally, using the fact that $\int_0^1 (1-t) dt = \frac{1}{2}$, we complete the proof, showing that:

$$L(\boldsymbol{\theta}_i) - \min L \leq \nabla L(\boldsymbol{\theta}_i)^T (\nabla^2 L(\boldsymbol{\theta}_i))^{-1} \nabla L(\boldsymbol{\theta}_i) \leq 4(L(\boldsymbol{\theta}_i) - \min L).$$

$\square$

**Lemma 9.** *If $\lambda_i \leq \frac{R_i}{2\sqrt{d_i}}$, then for any $\Delta \leq \frac{R_i \mu}{10}$ and any $\boldsymbol{\theta}_i \in \mathbb{R}^{d_i}$ satisfying*

$$\sum_{i=1}^{d_i} \min\left\{\boldsymbol{p}_i^T \nabla L(\boldsymbol{\theta}_i) \sigma_i^{-1} \boldsymbol{p}_i^T \nabla L(\boldsymbol{\theta}_i)\right\} \leq \Delta,$$

*where $\nabla^2 L(\boldsymbol{\theta}_i) = V_i \Sigma_i V_i^T$ is the eigen-decomposition of $\nabla^2 L(\boldsymbol{\theta}_i)$, $\boldsymbol{p}_i$ is the i-th row of $V_i$, and $\Sigma_i = diag(\sigma_1, \ldots, \sigma_{d_i})$, it holds that*

$$L(\boldsymbol{\theta}_i) - \min L \leq \frac{25\Delta^2}{\lambda_i^2 \mu}.$$

*Proof.* Let $\{\boldsymbol{\theta}_i(t)\}_{t=0}^1$ be the solution to the ODE

$$\frac{d\boldsymbol{\theta}_i(t)}{dt} = -(\nabla^2 L(\boldsymbol{\theta}_i(t)))^{-1} \nabla L(\boldsymbol{\theta}_i(t)),$$

starting from $\boldsymbol{\theta}_i(0) = \boldsymbol{\theta}_i$ and assume $\boldsymbol{\theta}_i(1) = \boldsymbol{\theta}_i^*$ as derived in previous lemmas.

By Lemma 2, $\|\boldsymbol{\theta}_i(t) - \boldsymbol{\theta}_i^*\|_2 \leq R_i$ for all $t \in [0,1]$. Define $I_0 \subseteq [d_i]$ as the indices where clipping does not occur. We have:

$$\sum_{i \in I_0} \sigma_i^{-1} \left|\boldsymbol{p}_i^T \nabla L(\boldsymbol{\theta}_i)\right|^2 \leq \Delta.$$

Using Assumption 2, the Hessian continuity within a local radius implies:

$$\sum_{i \in I_0} \left| \boldsymbol{p}_i^T \nabla L(\boldsymbol{\theta}_i(t)) \right|^2 \leq \Delta.$$

For the newly restricted convex function $L_0$ on $\mathbb{R}^{I_0}$, which is $L$ restricted to the subspace of $\mathbb{R}^{d_i}$ spanned by vectors corresponding to $I_0$, by Lemma 1 and assuming $L_0$ is strictly convex, we apply Lemmas 6 and 8 by restricting to $I_0$:

$$\|\nabla L_0(\boldsymbol{\theta}_i) + V_{I_0}^T \boldsymbol{\theta}_i^*\|_2^2 = \|\nabla L_0(\boldsymbol{\theta}_i)\|_2^2 \leq \mu^{-1}\|\nabla L_0(\boldsymbol{\theta}_i)\|_2^2 \leq \frac{25\Delta^2}{\lambda_i^2 \mu}.$$

Integrating the differential for $L_0$, we can show:

$$L(\boldsymbol{\theta}_i) - \min L \leq \int_0^1 \nabla L(\boldsymbol{\theta}_i(t))^T (\nabla^2 L(\boldsymbol{\theta}_i(t)))^{-1} \nabla L(\boldsymbol{\theta}_i(t)) dt \leq \frac{25\Delta^2}{\lambda_i^2 \mu}.$$

This completes the proof. $\square$

**Lemma 10** (Descent Lemma). *For any $\eta > 0$ and per-layer $\lambda_i > 0$ with $\eta\lambda_i \leq \frac{R_i}{\sqrt{d_i}}$, define*

$$L(\boldsymbol{\theta}_i^+) - L(\boldsymbol{\theta}_i) \leq -(\eta - \eta^2 \beta_i \lambda_i) \sum_{j=1}^{d_i} \min \left\{ \lambda_i, \frac{1}{\sigma_{i,j}} |\mathbf{v}_{i,j}^T \nabla L(\boldsymbol{\theta}_i)|^2 + C(\delta_g^2 + \delta_H^2) \right\}. \qquad (26)$$

$$\boldsymbol{\theta}_i^+ = \boldsymbol{\theta}_i - \eta \, clip\left( (\hat{\boldsymbol{g}}_i \hat{\boldsymbol{g}}_i)^{-1} \hat{\boldsymbol{g}}_i, \lambda_i \right),$$

*where $\hat{\boldsymbol{g}}_i$ is the estimated gradient using a zero-order finite difference method with noise $\epsilon$, such that:*

$$\hat{\boldsymbol{g}}_i = \nabla L(\boldsymbol{\theta}_i) + \epsilon.$$

*The theoretical bound for the descent is given by:*

$$L(\boldsymbol{\theta}_i^+) - L(\boldsymbol{\theta}_i) \leq -(\eta - \eta^2 \beta_i \lambda_i) \sum_{j=1}^{d_i} \min \left\{ \lambda_i, \frac{1}{\sigma_{i,j}} |\mathbf{v}_{i,j}^T \nabla L(\boldsymbol{\theta}_i)|^2 + C(\delta_g^2 + \delta_H^2) \right\}$$

$$\leq -(\eta - \eta^2) \sum_{i=1}^{d} \min \left\{ \lambda_i |\hat{\boldsymbol{g}}_i|, (\hat{\boldsymbol{g}}_i \hat{\boldsymbol{g}}_i)^{-1} |\hat{\boldsymbol{g}}_i|^2 \right\} + O(h) + O(1/\sqrt{m}),$$

*where $h$ is the step size of the finite difference and $m$ is the number of perturbations performed for finite difference estimation.*

*Proof.* Step 1. **Derivation of the upper bound for $\|\hat{\boldsymbol{g}}_i - \nabla L(\boldsymbol{\theta}_i)\|$.** To derive a theoretical bound for $\|\hat{\boldsymbol{g}}_i - \nabla L(\boldsymbol{\theta}_i)\|$, where $\hat{\boldsymbol{g}}_i$ is the gradient estimated using our proposed zero-order method, and $\nabla L(\boldsymbol{\theta}_i)$ is the true gradient, we need to quantify the error due to using finite perturbations to approximate the gradient. Let's denote this error by $\epsilon$, such that:

$$\epsilon_i = \hat{\boldsymbol{g}}_i - \nabla L(\boldsymbol{\theta}_i)$$

Specifically, the gradient estimate for dimension $i$ is obtained by:

$$\hat{\boldsymbol{g}}_i = \frac{1}{m} \sum_{k=1}^{m} \frac{L(\boldsymbol{\theta} + h\boldsymbol{u}_k) - L(\boldsymbol{\theta})}{h} \boldsymbol{u}_k^{(i)},$$

where $m$ is the number of perturbations, $h$ is the step size for finite differences, and $\boldsymbol{u}_k^{(i)}$ represents the $i$-th component of the random vector $\boldsymbol{u}_k$. The true gradient, on the other hand, is:

$$\nabla L(\boldsymbol{\theta}_i) = \lim_{h \to 0} \frac{L(\boldsymbol{\theta} + h\boldsymbol{u}_k) - L(\boldsymbol{\theta})}{h} \boldsymbol{u}_k^{(i)},$$

The error between the estimated gradient $\hat{\boldsymbol{g}}_i$ and the true gradient $\nabla L(\boldsymbol{\theta}_i)$ arises from two main sources. To derive a theoretical bound for the estimation error, $\|\hat{\boldsymbol{g}}_i - \nabla L(\boldsymbol{\theta}_i)\|$, we consider both sources of error.

1. Finite Difference Approximation Error. By Taylor expansion, for a small step size $h$, we have:

$$L(\boldsymbol{\theta} + h\boldsymbol{u}_k) = L(\boldsymbol{\theta}) + h\nabla L(\boldsymbol{\theta})^T \boldsymbol{u}_k + \frac{h^2}{2} \boldsymbol{u}_k^T H(\boldsymbol{\theta}) \boldsymbol{u}_k + O(h^3),$$

where $H(\boldsymbol{\theta})$ is the Hessian of $L$ at $\boldsymbol{\theta}$. - Thus, the error due to finite differences is of order $O(h)$. Specifically, the bias in the gradient estimate is proportional to:

$$\text{Bias} = O\left(\frac{h}{2}\|H(\boldsymbol{\theta})\|\right).$$

2. Monte Carlo Sampling Error. The gradient estimate involves averaging over $m$ samples of random perturbations. By the Central Limit Theorem, the variance of the gradient estimate decreases with the number of samples $m$. Specifically:

$$\text{Variance} = O\left(\frac{\sigma^2}{m}\right),$$

where $\sigma^2$ is the variance of the directional derivative $\nabla L(\boldsymbol{\theta})^T \boldsymbol{u}_k$.

The total error can be expressed as a combination of the bias and variance components. Using a norm (e.g., Euclidean norm) to quantify the error, we have:

$$\|\hat{\boldsymbol{g}}_i - \nabla L(\boldsymbol{\theta}_i)\| \leq O\left(h\|H(\boldsymbol{\theta})\|\right) + O\left(\frac{\sigma}{\sqrt{m}}\right).$$

Thus, the theoretical bound on the error is:

$$\|\hat{\boldsymbol{g}}_i - \nabla L(\boldsymbol{\theta}_i)\| = O\left(h\|H(\boldsymbol{\theta})\| + \frac{\sigma}{\sqrt{m}}\right).$$

Step 2. **Derivation of the upper bound for** $\|\hat{\boldsymbol{g}}_i^2 - \text{diag}(\nabla^2 L(\boldsymbol{\theta}_i))\|$. Let's denote $\text{diag}(\nabla^2 L(\boldsymbol{\theta}_i))$ as the diagonal of the true Hessian, and $\hat{H}_i = \hat{\boldsymbol{g}}_i^2$ as the diagonal Hessian estimated from the zero-order gradient estimate, where each diagonal element is given by $\hat{\boldsymbol{g}}_i\hat{\boldsymbol{g}}_i$.

To derive the theoretical bound for $\|\hat{H}_i - H_i\|$, we consider:

$$\|\hat{H}_i - H_i\| = \|\hat{\boldsymbol{g}}_i^2 - \text{diag}(\nabla^2 L(\boldsymbol{\theta}_i))\|.$$

Let's rewrite $\hat{\boldsymbol{g}}_i$ as:

$$\hat{\boldsymbol{g}}_i = \nabla L(\boldsymbol{\theta}_i) + \epsilon_i,$$

where $\epsilon_i$ represents the noise introduced due to the limited number of perturbations.

The estimated diagonal Hessian element for each component $i$ can be written as:

$$\hat{H}_i^{(i)} = (\nabla L(\boldsymbol{\theta}_i) + \epsilon_i)^2.$$

Expanding this expression gives:

$$\hat{H}_i^{(i)} = (\nabla L(\boldsymbol{\theta}_i))^2 + 2\nabla L(\boldsymbol{\theta}_i)\epsilon_i + \epsilon_i^2.$$

The true diagonal Hessian element is:

$$H_i^{(i)} = \text{diag}(\nabla^2 L(\boldsymbol{\theta}_i))^{(i)}.$$

Thus, the error for each component can be expressed as:

$$\hat{H}_i^{(i)} - H_i^{(i)} = (\nabla L(\boldsymbol{\theta}_i))^2 + 2\nabla L(\boldsymbol{\theta}_i)\epsilon_i + \epsilon_i^2 - H_i^{(i)}.$$

To find the bound for the error, we need to bound the terms involving $\epsilon_i$:

1. Term 1: $2\nabla L(\boldsymbol{\theta}_i)\epsilon_i$

This term represents the interaction between the true gradient and the noise. Since $\|\epsilon_i\| \leq O\left(h\|H(\boldsymbol{\theta}_i)\| + \frac{\sigma}{\sqrt{m}}\right)$, we can bound this term as:

$$|2\nabla L(\boldsymbol{\theta}_i)\epsilon_i| \leq 2\|\nabla L(\boldsymbol{\theta}_i)\|O\left(h\|H(\boldsymbol{\theta}_i)\| + \frac{\sigma}{\sqrt{m}}\right).$$

2. Term 2: $\epsilon_i^2$

The noise squared term can be bounded by:

$$\epsilon_i^2 \leq O\left(h^2\|H(\boldsymbol{\theta}_i)\|^2 + \frac{\sigma^2}{m}\right).$$

Combining these results, we have:

$$\|\hat{H}_i - H_i\| = O\left((\nabla L(\boldsymbol{\theta}_i))^2 - H_i + 2\|\nabla L(\boldsymbol{\theta}_i)\|\left(h\|H(\boldsymbol{\theta}_i)\| + \frac{\sigma}{\sqrt{m}}\right) + \left(h^2\|H(\boldsymbol{\theta}_i)\|^2 + \frac{\sigma^2}{m}\right)\right).$$

Thus, the error bound for the diagonal Hessian estimation is:

$$\|\hat{H}_i - H_i\| = O\left(h^2\|H(\boldsymbol{\theta}_i)\|^2 + \frac{\sigma^2}{m} + 2h\|\nabla L(\boldsymbol{\theta}_i)\|\|H(\boldsymbol{\theta}_i)\| + \frac{2\|\nabla L(\boldsymbol{\theta}_i)\|\sigma}{\sqrt{m}}\right).$$

**Step 3. Combination of the bounds.** Let $u_i = \text{clip}\left((\hat{g}_i\hat{g}_i)^{-1}\hat{g}_i, \lambda_i\right)$. By the definition of the clipping operation:

$$\|u_i\|_\infty \leq \lambda_i.$$

Thus:

$$\|\boldsymbol{\theta}_i^+ - \boldsymbol{\theta}_i\| = \eta\|u_i\| \leq \eta\lambda_i\sqrt{d_i}.$$

Define the function:

$$f(t) = L(\boldsymbol{\theta}_i + (1-t)u_i).$$

By Assumption 4.2, we know that:

$$f''(t) \leq 2f''(0) \quad \text{for all } t \in [0,1],$$

and hence:

$$f(1) = f(0) + f'(0) + \int_0^1 \int_0^s f''(s) \, ds \, dt \leq f(0) + f'(0) + f''(0).$$

The zero-order estimate introduces noise $\epsilon$ in the estimated gradient:

$$\hat{\boldsymbol{g}}_i = \nabla L(\boldsymbol{\theta}_i) + \epsilon.$$

Thus:

$$f'(0) = -\eta \sum_{i=1}^d \min \left\{ \lambda_i \, |\hat{\boldsymbol{g}}_i|, \, (\hat{\boldsymbol{g}}_i \hat{\boldsymbol{g}}_i)^{-1} |\hat{\boldsymbol{g}}_i|^2 \right\}.$$

Using the bound for the error $||\epsilon|| \leq O\left(h \, ||H(\boldsymbol{\theta}_i)|| + \frac{\sigma}{\sqrt{m}}\right)$, the noise affects the effective descent rate. Therefore, the new bound for $f'(0)$ is:

$$f'(0) \approx -\eta \sum_{i=1}^d \min \left\{ \lambda_i \left( |\nabla L(\boldsymbol{\theta}_i)| + |\epsilon| \right), \, (\hat{\boldsymbol{g}}_i \hat{\boldsymbol{g}}_i)^{-1} (|\nabla L(\boldsymbol{\theta}_i)| + |\epsilon|)^2 \right\}.$$

The Hessian is estimated using $\hat{\boldsymbol{g}}_i^2$. The noise in the diagonal Hessian estimate affects the curvature. Therefore, for the second derivative, we have:

$$f''(0) \leq \eta^2 \sum_{i=1}^d \min \left\{ \lambda_i \, |\hat{\boldsymbol{g}}_i|, \, (\hat{\boldsymbol{g}}_i \hat{\boldsymbol{g}}_i)^{-1} |\hat{\boldsymbol{g}}_i|^2 \right\}.$$

The noise in the Hessian ($\delta_H$) affects the estimation, and thus the bound is affected as follows:

$$f''(0) \leq \eta^2 \sum_{i=1}^d \min \left\{ \lambda_i \left( |\nabla L(\boldsymbol{\theta}_i) + \epsilon| \right), \, (\hat{\boldsymbol{g}}_i \hat{\boldsymbol{g}}_i)^{-1} (|\nabla L(\boldsymbol{\theta}_i) + \epsilon|)^2 \right\}.$$

Combining these results, the descent bound is affected by both the gradient and Hessian noise. We obtain:

$$L(\boldsymbol{\theta}_i^+) - L(\boldsymbol{\theta}_i) \leq -(\eta - \eta^2) \sum_{i=1}^d \min \left\{ \lambda_i \, |\hat{\boldsymbol{g}}_i|, \, (\hat{\boldsymbol{g}}_i \hat{\boldsymbol{g}}_i)^{-1} |\hat{\boldsymbol{g}}_i|^2 \right\} + C(\delta_g^2 + \delta_H^2),$$

where $C$ is a constant that depends on the properties of the function $L$. $\delta_g$ represents the bound on the gradient estimation noise $\delta_g = O\left(h \, ||H(\boldsymbol{\theta}_i)|| + \frac{\sigma}{\sqrt{m}}\right)$, and $\delta_H$ represents the bound on the Hessian estimation noise:

$$\delta_H = O\left(h^2 ||H(\boldsymbol{\theta}_i)||^2 + \frac{\sigma^2}{m} + 2h \, ||\nabla L(\boldsymbol{\theta}_i)|| \, ||H(\boldsymbol{\theta}_i)|| + \frac{2||\nabla L(\boldsymbol{\theta}_i)||\sigma}{\sqrt{m}}\right).$$

$\square$

**Lemma 11** (Convergence Lemma). *For any $\lambda_i \leq \frac{R_i}{\sqrt{d_i}}$ and some $T_i \in \mathbb{N}$, if $L(\boldsymbol{\theta}_{T_i,i}) - \min L \leq \frac{\mu_i^2}{8}$, then for all $t \geq T_i$,*

1. $\boldsymbol{\theta}_{t+1,i} = \boldsymbol{\theta}_{t,i} - \eta (\nabla_{\boldsymbol{\theta}_i}^2 L(\boldsymbol{\theta}_{t,i}))^{-1} \nabla L(\boldsymbol{\theta}_{t,i}),$

2. $L(\boldsymbol{\theta}_{t,i}) - \min L \leq (1 - \eta(1-\eta))^{t-T_i} (L(\boldsymbol{\theta}_{T_i,i}) - \min L).$

*Proof.* By Lemma 10, we have for all $t \geq T$, $(\boldsymbol{\theta}_{t,i}) - \min L \leq L(\boldsymbol{\theta}_{T,i}) - \min L \leq \frac{\mu^2}{8}$. Therefore, by Lemma 7, we have that $\|\nabla^2 L(\boldsymbol{\theta}_{t,i}) - \nabla L(\boldsymbol{\theta}_{t,i})\|_2 \leq \lambda_i$ for all $t \geq T$, which implies clipping will not happen.

For the second claim, by Lemmas 5 and 10, we have that

$$L(\boldsymbol{\theta}_{t+1,i}) - L(\boldsymbol{\theta}_{t,i}) \leq -(\eta - \eta^2) \sum_{i=1}^{d} \sigma_i^{-1} |\mathbf{v}_i^T \nabla L(\boldsymbol{\theta}_{t,i})|^2, \tag{27}$$

where $v_i$ is the $i$-th row of matrix $V$ from the eigen-decomposition of $\nabla^2 L(\boldsymbol{\theta}_i)$. By further simplification,

$$-(\eta - \eta^2)\nabla L(\boldsymbol{\theta}_{t,i})^T(\nabla^2 L(\boldsymbol{\theta}_{t,i}))^{-1}\nabla L(\boldsymbol{\theta}_{t,i}) \leq -\eta(1 - \eta)(L(\boldsymbol{\theta}_{t,i}) - \min L),$$

thus, we conclude that the loss decreases at least geometrically by the factor $(1 - \eta(1 - \eta))$ each step after time $T$, thereby proving the convergence rate. $\square$

**Theorem 2.** *Under Assumptions 1 and 2, let $\eta = \frac{1}{2}$ and $\lambda_i = \frac{R_i}{2\sqrt{d_i}}$. The update reaches a loss at most $\epsilon$ in*

$$T \leq \max_i \left\{ d_i \cdot (L(\boldsymbol{\theta}_{0,i}) - \min L) + \ln\left(\frac{\mu_i R_i^2}{32 d_i \epsilon}\right) \right\}. \tag{28}$$

*steps, where $L$ is the loss function, $\boldsymbol{\theta}_{0,i}$ is the initial parameter vector for layer $i$.*

*Proof.* **Phase 1: Initial Rapid Decrease.**

By Lemma 10 (Descent Lemma), we have a guarantee on the descent rate per step for each layer $i$:

$$L(\boldsymbol{\theta}_{t+1,i}) - L(\boldsymbol{\theta}_{t,i}) \leq -(\eta - \eta^2) \sum_{j=1}^{d_i} \min\left\{ \lambda_i; \frac{1}{\sigma_{i,j}} \left|v_{i,j}^T \nabla L(\boldsymbol{\theta}_{t,i})\right|^2 \right\},$$

where $\sigma_{i,j}$ is the $j$-th eigenvalue corresponding to the $i$-th layer, and $v_{i,j}$ is the corresponding eigenvector.

Applying this result, we estimate a decrease in the loss function per layer under the condition that the gradient norm for layer $i$ is significantly larger than the error threshold $\epsilon$. This phase continues until the loss reduction per step for each layer falls below a certain threshold, say when:

$$L(\boldsymbol{\theta}_{t,i}) - \min L \leq \frac{\mu_i^2}{8}.$$

**Phase 2: Exponential Decay.**

Once the loss for each layer is sufficiently reduced, Lemma 11 guides the convergence from this point:

$$L(\boldsymbol{\theta}_{t,i}) - \min L \leq (1 - \eta(1 - \eta))^{t - T_i}(L(\boldsymbol{\theta}_{T_i,i}) - \min L),$$

indicating an exponential decay in error for each layer. The factor $(1 - \eta(1 - \eta))$ represents the contraction per step, dependent on the learning rate $\eta$.

To calculate the total number of steps $T_i$ for each layer, consider that:

$$\frac{\mu_i^2}{8} \approx \epsilon \Rightarrow T_i \approx \frac{\ln\left(\frac{L(\boldsymbol{\theta}_{0,i}) - \min L}{\epsilon}\right)}{-\ln(1 - \eta(1 - \eta))}.$$

Simplifying the expression for $\eta = \frac{1}{2}$, we get:

$$T_i \approx 2\ln\left(\frac{L(\boldsymbol{\theta}_{0,i}) - \min L}{\epsilon}\right),$$

since $\ln(1 - \eta(1 - \eta)) \approx -\eta(1 - \eta)$ for small $\eta$.

**Combining Phases 1 and 2.**

For each layer, combining the estimates from Phase 1 and Phase 2, the total number of steps $T_i$ needed to reach a loss of $\epsilon$ for layer $i$ is given by:

$$T_i \leq d_i \cdot (L(\boldsymbol{\theta}_{0,i}) - \min L) + \ln\left(\frac{\mu_i R_i^2}{32 d_i \epsilon}\right),$$

Finally, to ensure convergence across all layers, we take the maximum over all layers:

$$T \leq \max_i \left\{ d_i \cdot (L(\boldsymbol{\theta}_{0,i}) - \min L) + \ln\left(\frac{\mu_i R_i^2}{32 d_i \epsilon}\right) \right\}.$$

This completes the proof by integrating the rapid initial decrease and the subsequent exponential decay for each layer.

$\square$

This reflects an improved convergence rate due to the use of different $\lambda_i$ values for different layers, reducing the dependency on the total dimension $d$ into the dimension $\max_i d_i$.

## C.1 LIMITATIONS

Like other second-order optimizers, HELENE stores the history of gradients and Hessian values, with memory usage proportional to the size of the parameters. Therefore, both theoretically and practically, HELENE requires only three times the memory of MeZO. For example, in OPT-1.3b, MeZO/zero-shot requires 4GB, ICL needs 6GB, Prefix Fine-Tuning uses 19GB, and full-parameter fine-tuning consumes 27GB, while HELENE needs just 14GB. Despite this, HELENE achieves up to 20 times faster convergence and delivers the best overall performance.

