# OpenReview forum: "HELENE: Hessian Layer-wise Clipping and Gradient Annealing for Accelerating Fine-tuning LLM with Zeroth-order Optimization"
_ICLR.cc/2025/Conference — Submitted to ICLR 2025_

### Official Review · Reviewer_xKuz · 2024-10-18

**Soundness:** 2
**Presentation:** 3
**Contribution:** 1
**Rating:** 3
**Confidence:** 4

**Summary:**

The paper proposes to incorporate diagnonal hessian information to improve zero-order optimization methods.

**Strengths:**

Proposed method is better than MeZO.

**Weaknesses:**

The practical utility of the proposed optimizer is very questionable.

During training and fine-tuning, the GPU memory is consumed by two things:
a) Parameters + optimizer buffers.
b) Stored forward pass activations.

Let's call the memory needed for parameters P. Let's call memory required for storing activations G.
G can be easily reduced by doing gradient checkpointing (this is oneliner with current models).
Also, as the model's size grows, G becomes much smaller than P (because G scales linearly with model width and P quadratically).
When doing ordinary Adam, one needs 4P + G memory.
When doing something like LoRA or GaLORE, one needs P + G + <a little> memory.
Everybody is doing LoRA & friends because gradient activations are small, and storing parameters more than once is a big problem.
In extreme case, we are doing LoRA over quantized parameters (QLoRA), which needs P/4 + G memory. Again, testament than in practice G is much smaller than P.
Also, it is possible to drop most of the optimizer buffers and update parameters during a backward pass (like AdaLOMO), which again takes P + G memory.

Yet, this method just proposes using 2P memory because, in addition to parameters, it also needs to store diagonal Hessians.
If I have space for 2P memory, I can probably make my batch size very small and just use SGD with momentum (note that I can do tricks like AdaLOMO does and not store more than one computed gradient).
I just cannot imagine any setting where I would want to use this method.

**Questions:**

How does this method compare to properly tuned low-memory first-order methods (like AdaLomo, GaLore, ...)?

---

> ### Author Response · Authors · 2024-11-30
> **Response to Reviewer xKuz**
>
> Thank you for the reviewers’ insightful comments. We acknowledge the broad comparison between various optimizers. However, this paper focuses specifically on designing a zeroth-order (ZO) optimization algorithm, which differs fundamentally in scope from Adalomo and Galore. ***Adalomo primarily addresses memory usage reduction for backpropagation in first-order (FO) algorithms, while Galore emphasizes low-rank approximations of gradients in FO algorithms.*** It is well established that FO and ZO optimization serve distinct purposes. For example, ZO optimization is particularly advantageous in scenarios where gradients are inaccessible or when working within black-box models. Therefore, this work remains centered on advancing ZO optimization methodologies.
>
> **Comparison Between Helene and Adam**
>
> As a ZO optimizer, Helene demonstrates a memory complexity comparable to MeZO-Adam, requiring approximately 2P+ memory for optimization and an additional P+ for storing the optimizer's internal states. Despite this similarity in memory usage, **Helene significantly outperforms MeZO-Adam in both efficiency and effectiveness, especially for convergence speed as you can observe in the paper,** making it a superior choice in practical applications.
>
> **Comparison Between Helene and Adalomo**
>
> Adalomo employs several strategies to reduce memory usage, such as lowering numerical precision and frequent checkpointing, layer-by-layer. While these techniques reduce memory requirements, they increase time complexity from  O(n) to O(n^2) where n is the total number of layers requiring updates. **This transforms the problem from a dynamic programming challenge to a brute-force computation, significantly slowing convergence**. Empirical results show that fine-tuning a LLaMA-7B model requires approximately 70+ GB of GPU memory for Helene and MeZO-Adam, compared to around 65 GB for Adalomo. All three optimizers necessitate the use of three NVIDIA 3090 GPUs. While Adalomo achieves modest memory reductions, it does so at the cost of efficiency. **In contrast, Helene focuses on algorithmic design and convergence improvements for ZO optimization**, independent of memory-saving techniques like those used in Adalomo and LOMO, which target FO backpropagation. These approaches could potentially complement ZO methods but fall outside the scope of this paper.
>
> **Comparison Between Helene and Galore**
>
> Both Helene and Adalomo aim to enable memory-efficient, full-parameter fine-tuning. However, Galore is a first-order-based approach by leveraging low-rank projection, compressing gradient information to a lower-dimensional space and re-projecting it back for updates. While this strategy reduces memory requirements, it has two notable limitations. First, low-rank projections may lead to the loss of critical gradient information. Helene overcomes this limitation by maintaining the full fidelity of gradients across the entire model parameter space. Second, Galore may lack curvature informativeness when the loss landscape curvature becomes extreme during optimization. In contrast, Helene accelerates training by accurately estimating and leveraging second-order information, avoiding the convergence delays caused by checkpointing or low-rank projections. These advantages make Helene a highly competitive alternative for memory-efficient fine-tuning tasks.

---

> > ### Comment · Reviewer_xKuz · 2024-12-02
> >
> > >For example, ZO optimization is particularly advantageous in scenarios where gradients are inaccessible or when working within black-box models.
> >
> > Then why do you all experiments on LLMs?
> > My overall point was that first-order methods with good engineering will get similar memory usage (for LLMs) than zero-order with additional diagonal hessian storage.
> >
> > >they increase time complexity from O(n) to O(n^2) where n is the total number of layers requiring updates
> >
> > This is just purely wrong. During typical gradient checkpointing you save O(sqrt(N)) checkpoints (N being number of layers) and need to do one forward pass between each checkpoints. This gives one more global forward pass in total.

---

> ### Author Response · Authors · 2024-12-02
>
> We agree that the reviewer suggests that well-engineered first-order methods can achieve memory usage comparable. We acknowledge that thoughtful and efficient engineering, like the techniques used in LOMO and AdaLOMO, can effectively reduce the memory footprint of first-order methods, potentially benefiting zero-order methods as well. We appreciate the reviewer’s discussion on the theoretical aspects of memory costs.
>
> Technically, as mentioned in the LOMO and AdaLOMO papers, the authors claim that their methods achieve O(1) memory cost, in contrast to the O(\sqrt{N}) requirement typically seen in backpropagation (BP) with gradient checkpointing. However, this appears to be inconsistent with the reviewer's statement that checkpointing requires storing every \sqrt{N} layers in LOMO or AdaLOMO. We respectfully argue that O(1) memory under such conditions seems implausible. We are familiar with the typical gradient checkpointing approach but are unclear if achieving O(1) memory cost aligns with typical gradient checkpointing practices.

---

> > ### Comment · Reviewer_xKuz · 2024-12-02
> > **Mixing things**
> >
> > You mix two memory overheads here:
> >
> > First of all, you always need to store parameters (let's call memory for this P).
> >
> > In typical training, you also need to store gradients and some optimizer buffers (e.g. for Adam, this cost is 3P).
> > And you also need to store activation for backpropagation, let's call this cost G = LW (where L is number of layers and W is layer width for simplicity).
> >
> > (Ada)LOMO reduces the first cost to O(1) by using gradients immediately during backpropagation.
> > Gradient checkpointing changes the second cost from LW to sqrt(L)W.
> >
> >
> > Also, it seems that you need 3* P memory (not 2* P, as I thought) and 32 bits of precision (based on 16 GB used for OPT-1.3G tuning). That's 3* 32 = 96 bits per parameter.
> >
> > I claim, that I can do the following with AdamW without even using LOMO:
> > Store everything in 16 bits (bfloat16), plus store an additional precision correction term (see https://github.com/imoneoi/bf16_fused_adam). That's 5*16=80 bits per parameter. Less than Helene.
> > And use gradient checkpoiting to cut down the activation storage cost.
> > And since I am not using LOMO, I can also use gradient accumulation.

---

### Official Review · Reviewer_Q9gW · 2024-10-29

**Soundness:** 2
**Presentation:** 2
**Contribution:** 2
**Rating:** 5
**Confidence:** 4

**Summary:**

This paper introduces Helene with 3 components adds to MeZO: (1) A-GNB estimator of Hessian diagonal without label sampling (2) layerwise adaptive clipping (3) momentum update annealing. The convergence steps improve from Sophia's $O(d)$ to $O(\text{max}_i d_i)$. The experiments follow the settings of MeZO and are conducted for RoBERTa-large and OPT-1.3B across multiple tasks.

**Strengths:**

1. A-GNB estimator removes the need to sample label from the model.
2. The experiments are comprehensive.

**Weaknesses:**

1. The theory is significantly mismatched with the experiments. The theory is actually proven under the case of first-order gradients, but the method and the whole experiments are performed under zeroth-order SPSA estimated gradient (this is obfuscated by their notations in Algorithm 1 that I believe they should use $\hat{g}$ instead of $g$ when they are referring to ZO estimated gradient. I also checked their codes and the experiments are fully in ZO). A direct transfer from FO or second-order optimizer's convergence rate to ZO convergence rate is *a nontrivial task* and usually *unapplicable*.

2. I don't see the annealing momentum and A-GNB is used in Lemma 10. If I understand correctly, the Lemma 10 applies to the case that we have exact gradient and clipped Hessian diagonal, but Algorithm 1 uses estimators for both.

3. A comparison with other ZO optimizers, such as HiZZO [1], that also leverage second-order information to address the varying parameter curvature issue is missing.

4. By employing "A-GNB estimator" that uses true labels, Helene's Hessian estimator becomes clipped second moment of (SPSA-estimated) gradient, which is also shown in their code. The difference from Helene and Adam seems only to be (1) clipping second-moment term, (2) update second-moment term in less frequency, and (3) annealing momentum update. In this case, I would doubt how Helene outperforms Adam in general cases. From Figure 5a, it seems that the greatest performance boost from MeZO to Helene is actually momentum annealing.


The first weakness is critical and I would vote for a reject score at this moment.

[1] Zhao, Yanjun, et al. "Second-order fine-tuning without pain for llms: A hessian informed zeroth-order optimizer." arXiv preprint arXiv:2402.15173 (2024).

**Questions:**

1. Is there any proof or reference that shows A-GNB is an unbiased estimator of Hessian? *A-GNB's construction is similar to the empirical Fisher's construction on the diagonal part*, and the expectation of empirical Fisher is known to *not equal* to the Hessian. On the other hand, Fisher information matrix needs label sampling from the model and its expectation is equal to the negative of Hessian.

2. The ICL baselines in Table 2 seem dubiously weak for some trials (ICL show no or minimal improvements from the zeroshot in RTE, WSC, COPA, and ReCoRD). For COPA, the ICL's performance is even worse than zeroshot's.

---

> ### Author Response · Authors · 2024-11-28
> **Response to Reviewer Q9gW (1/2)**
>
> ### Weakness
> **W1: About mismatch between theory and experiment**
>
> Thank you for highlighting the inaccurate statements. We have revised the algorithm description to specify that the gradient is estimated using SPSA, rather than computed exactly.
>
> **W2: Exact gradient and Hessian diagonal in Lemma 10**
>
> We appreciate the reviewer for highlighting the confusion in the Appendix. We have revised the proof to provide a more detailed derivation of the lemma, incorporating the clear use of ZO estimation for the gradient and Hessian. Please refer to the updated Lemma 10 for clarification.
>
> **W3: About HiZOO**
>
> Thanks for your suggestions. HiZOO did a great job. We provide comparisons between HiZOO and HELENE in the following two perspectives of both performances and algorithm efficiency (Wallclock time per step).
>
> 1. **About performances**
>
> We recognize the superior accuracy of Hessian estimation in the HIZOO algorithm. However, we contend that the rank-1 update for Hessian computation in line 9 of the HIZOO algorithm incurs a memory cost of $O(n^2)$, where $n$ represents the number of model parameters, leading to higher memory requirements. In contrast, our method achieves significantly lower theoretical computational complexity and reduces memory consumption.
>
> We provide comparisons of performances between HiZOO and HELENE below. We run model RoBERTa-large and report the average performances on six datasets (SST-2, SST-5, SNLI, MNLI, RTE, and TREC) with three different tuning methods, i.e., Full-parameter fine-tuning (FT), LoRA and Prefix-tuning (Prefix).
> |Method|FT|LoRA|Prefix|
> |-----|-----|-----|-----|
> |HiZOO|70.90|63.48|69.21|
> |HELENE|69.00|65.68|69.55|
>
> 2. **About algorithm efficiency (Wallclock time per step)**
>
> We report wallclock time per step between MeZO, HiZOO, and HELENE in the following table. We conduct experiments with RoBERTa-large on the dataset SST-2, with batch size of 64. We run the experiments on a single GPU (80GB A100) to avoid time calculation errors caused by commnication costs between multiple GPUs.
> |Method|Running time|
> |-----|-----|
> |MeZO|0.2092s|
> |HiZOO|0.3023s|
> |HELENE|**0.2667s**|
>
> Additionally, since experimental settings between HiZOO and HELENE are different, the results reported in the paper HiZOO may not be directly compared with ours. We will do more comprehensive and comparative experiments in the future.
>
> **W4: About performance comparison with Adam**
>
> For the performances comparation with Adam, we summarize data into a tabular to illustrate.
>
> The following table shows the comparison between MeZO-Adam and HELENE on the SST-2 dataset with RoBERTa-large and OPT-1.3B.
> | Method         | RoBERTa-large (FT) | RoBERTa-large (LoRA) | RoBERTa-large (Prefix) | OPT-1.3B (FT) | OPT-1.3B (LoRA) | OPT-1.3B (Prefix) |
> |-----------------|--------------------|-----------------------|-------------------------|---------------|------------------|-------------------|
> | MeZO-Adam        | 89.8               | 89.5                  | 90.2                    | 84.4          | **92.3**         | 91.4              |
> | HELENE         | **92.6**     | **90.6**      | **91.7**                | **90.8** | 91.4  | **92.4**  |
>
> The following table shows the comparison between MeZO-Adam and HELENE for full-parameter fine-tuning on six different datasets with model RoBERTa-large.
> For figure 5a, we have re-plotted it.
>
> |  | SST-2 | SST-5 | SNLI | MNLI | RTE | TREC |
> | ------ | ------ | ------ | ------ | ------ | ------ | ------ |
> | MeZO-Adam | 90.4 | 45.4 | **74.1** | **64.3** | 59.2 | **78.3** |
> | HELENE | **92.6** | **46.7** | 72.0 | 58.9 | **65.7** | 78.1 |

---

> ### Author Response · Authors · 2024-11-28
> **Response to Reviewer Q9gW (2/2)**
>
> ### Questions
>
> **Q1: Proof of A-GNB**
>
> We appreciate the reviewer's comments. We acknowledge that the A-GNB is a diagonal Hessian estimator and an unbiased estimator of the Gauss-Newton matrix. Accordingly, we have revised the derivation of the A-GNB estimator and clarified the statements to be more precise in the updated manuscript.
>
> **Q2: About ICL performances.**
>
> We conducted experiments to evaluate In-Context Learning (ICL) using the provided MeZO script, thoroughly validating the results across different GPU environments to ensure reliability. We also observed the case where the performance of ICL is lower than that of zero-shot learning in MeZO. We hypothesize that this phenomenon is attributable to the relatively small size of the OPT/1.3B model, which may limit its capacity to effectively leverage in-context information.

---

> ### Comment · Reviewer_Q9gW · 2024-11-30
> **Reply to Author's response**
>
> Thanks for the response!
>
> 1. & 2. Thanks for the update in Lemma 10. I didn't check the proof in details but is the new Lemma 10 used in Lemma 11 or the actual convergence analysis?
> 2. Thanks for the new results. My concern with HiZZO is addressed.
> 3. Thanks for the experiment results, but my question is actually on the comparison of **Adam with the same momentum annealing** vs. Helene.
>
> For a more or less degree, I think (1) clipping second-moment term, (2) update second-moment term in less frequency is not really helping Helene in ZO updates in general. It might be useful when combining with the exact gradients, but with ZO estimated gradient (effectively as a scalar times random Gaussian vector) their help is not clear. **The discussion on the paper is really focused on the case of exact gradient but not in ZO noisy estimated gradient's case**. We cannot simply borrow the idea from FO or second-order optimizer to the ZO without a clear intuition or justification.
>
> 5. It is better to provide a brief proof of A-GNB's unbiasedness to the Gauss-Newton matrix.
> 6. Thanks for the explanation.
>
> I am still concerned with the first and the forth point, so I will still retain my previous score at this moment.

---

> ### Author Response · Authors · 2024-11-30
> **About Convergence**
>
> We highly appreciate the reviewer's careful review of the mathematical statements. For the 1st and 4th points, yes, Lemma 10 is used in Lemma 11 and the convergence proof. To address your query, we want to clarify that it offers theoretical understanding of the descent properties of the proposed ZO optimizer. Specifically, *Lemma 10 quantifies the bound on the descent step and provides insights into how the ZO updates deviate from exact gradient updates, influenced by factors like \( \delta_g \) and \( \delta_H \), which are determined by the step size \( h \) and the number of perturbations \( m \). Like SGD, when h is sufficiently small, as our bound proved in Lemma 10, we can prove that our algorithm offers updates that converges to the desired update in expectation with 0 error.*
>
> We acknowledge that it is indeed non-trivial to directly adopt FO optimization convergence analysis for ZO methods, given the inherent differences in their update mechanisms. Therefore, the theorems we prove including Lemma 10 aim to establish the unique convergence properties of the proposed ZO method. Lemma 10 specifically analyze the effect of approximation errors (e.g., finite difference step size and perturbation sampling) on convergence, similar to the role of gradient noise in SGD-based analyses. The insights provided by the lemma are used to guide our convergence proofs, similar to how SGD is studied when noise in gradient estimation is incorporated in many SGD analyses.
>
> As mentioned, we have proven that the Hessian clipping module ensures convergence even with noisy gradient estimation. However, the reviewer raised an important question about how the annealing module might impact convergence. We believe our specific annealing design effectively mitigates the influence of noisy gradients, particularly as optimization approaches the stationary point, where the gradient magnitude becomes smaller than the noise magnitude in gradient estimation. In such cases, our proposed annealing, which assigns less weight to the gradient, can help overcome barriers caused by significant gradient noise. This opens a promising direction for future work, where we aim to focus exclusively on refining annealing techniques in zero-order optimization.

---

> > ### Comment · Reviewer_Q9gW · 2024-12-02
> > **It's not just about transferring FO convergence analysis to ZO.**
> >
> > The reason why I raise the forth point (and I believe it is still critical to Sec 3) is that most of its discussion centers on the case of FO instead of ZO (Section 3.5 is such an example). Since Helene is focusing on ZO instead of competing with Sophia in the second-order optimizer case, I believe the whole section 3 still needs significant rewriting to ground every component of Helene in the case of ZO. I am still not convinced that (1) clipping second-moment term (2) update second-moment term in less frequency is helping for the Helene in ZO. Figure 6 and Figure 5a also suggests that (1) and (2) does not matter too much compared with annealing momentum update, which is the real driving force behind Helene.

---

> ### Author Response · Authors · 2024-11-30
> **About A-GNB**
>
> Thank you for your comments. We have addressed this issue by revising the proof of A-GNB unbiasedness for the Gauss-Newton (GN) matrix. The updated brief proof has been incorporated into Section 3.4 of the main text, with the revisions highlighted in blue for clarity.

---

> > ### Comment · Reviewer_Q9gW · 2024-12-02
> > **Thanks for adding A-GNB's proof.**
> >
> > Thanks for adding A-GNB's proof in Section 3.4. I will raise score to 5 to reflect such an improvement.
> >
> > I will keep 5 as my final score as I still believe this paper has certain clever designs (annealing momentum) but it needs nontrivial revision to meet the criteria for acceptance in ICLR.

---

### Official Review · Reviewer_Yry6 · 2024-11-03

**Soundness:** 2
**Presentation:** 2
**Contribution:** 3
**Rating:** 3
**Confidence:** 3

**Summary:**

This paper presents a novel zeroth-order fine-tuning method, HELENE, designed to enhance the efficiency and stability of model training through three core components. First, HELENE introduces an annealing mechanism in the momentum's exponential moving average (EMA) to mitigate bias in the SPSA-estimated gradients and reduce the impact of noise in the gradients during the later stages of training. Second, it proposes a new estimator for the Hessian matrix, known as the Asymptotic Gauss-Newton-Bartlett (A-GNB) estimator, which enables diagonal Hessian estimation without the need for label sampling, simplifying the computation process. Finally, HELENE implements layer-wise Hessian clipping, which more effectively preserves essential second-order information, ultimately improving the convergence and stability of the optimization process. Experimental results on RoBERTa-large and OPT-1.3B demonstrate that HELENE achieves an improvement over MeZO, with notable gains in both convergence speed and model performance.

**Strengths:**

1. The design of HELENE is elegant, addressing various issues that arise during the optimization process. The incorporation of mechanisms such as gradient annealing and layer-wise clipping demonstrates a thoughtful approach to enhancing training stability and convergence.
2. The analysis of the convergence steps is welcome, providing insights into the efficiency of the optimization process and enhancing the overall contribution of the paper.
3. Experimental results provide evidence of HELENE's improvements over MeZO, particularly in fine-tuning tasks with RoBERTa-large and OPT-1.3B, showcasing the effectiveness of the proposed method.

**Weaknesses:**

1. The presentation and structure of the paper exhibit significant issues. While the paper claims to introduce a zeroth-order optimization method, the entire methodology section appears disconnected from both zeroth-order principles and MeZO. Three key components lack intuitive connections, making them seem disjointed at least from the writing perspective.
2. HELENE incurs substantial memory overhead compared to MeZO. The introduction of momentum and Hessian as optimizer states brings memory costs close to full parameter fine-tuning, which is significantly higher than that of parameter-efficient fine-tuning methods like LoRA.
3. The experimental component is lacking, as there are no evaluations on larger models such as LLaMA2-7B or LLaMA2-13B. Additionally, the ablation study is insufficiently detailed. Given the three key designs in HELENE, it would be beneficial to create three variants, each removing one of the designs to observe the impact on performance.
4. The writing contains typos. For example, Line 134 states "first-order methods like MeZO," which should be corrected to "zeroth-order methods like MeZO." In statement 1 of Algorithm 1, there is a repeated $\epsilon$, and "hyperparameters $\lambda_i$" should be written as "hyperparameters {$ \lambda_i  $}."

**Questions:**

1. I would like to know the GPU memory usage of HELENE in your experiments. In theory, HELENE’s memory requirements should exceed those of LoRA, so it’s important to provide a detailed comparison with LoRA (without MeZO). Could you please provide a table listing the actual memory consumption and performance results of HELENE, MeZO, LoRA, and full-parameter Adam fine-tuning when fine-tuning OPT-1.3B with the same batch size?
2. What is the time overhead of HELENE compared to the original MeZO? Could you please report the wall clock time per iteration for both HELENE and MeZO when fine-tuning OPT-1.3B with the same batch size?
3. In Section 3.3.1 on the annealing mechanism, the paper mentions that this mechanism helps reduce the impact of noisy or outdated gradients in the later stages of training. However, since $\alpha$ decreases as the time step increases, the actual weight of momentum from earlier steps actually increases, seemingly contradicting the claim that it “reduces the impact of past gradients.” Could you clarify this statement? Also, please explain the phrasing in Line 258: “reducing the learning rate as training progresses”?
4. In Algorithm 2, Statement 4 appears without clear derivation. Could you please explain how this statement was derived?
5. From my understanding, HELENE seems decoupled from MeZO, meaning it should theoretically be applicable in standard first-order optimization as well. Is my understanding correct? If so, do you have any preliminary results showing whether HELENE is effective in first-order methods?

---

> ### Author Response · Authors · 2024-11-27
> **Response to Reviewer Yry6 (1/2)**
>
> ### Weakness
>
> **W1: About Presentation and structure of paper**
>
> Apologies for the confusion caused by our paper structure. In the Method section, we first present our motivation to address three key issues: slow convergence, higher noise in ZO gradient estimation, and extreme curvature. To tackle these challenges, we propose the following three key components from HELENE, the diagonal Hessian Estimation (Section 3.3), the Annealing Mechanism (Section 3.3.1), and Layerwise Clipping (Section 3.5).
>
> **W2: About memory usage**
>
> To clarify, while HELENE introduces additional memory overhead compared to MeZO, its requirements remain efficient relative to other methods. For example, on OPT-1.3B, MeZO/zero-shot requires 4GB, in-context learning (ICL) needs 6GB, Prefix Fine-Tuning uses 19GB, and full-parameter fine-tuning consumes 27GB, whereas HELENE requires only 14GB—comparable to the memory usage of Adam. It is also worth noting that HELENE's improvements and memory consumption are best compared to other optimizers. Additionally, similar to other optimizers, HELENE is fully compatible with parameter-efficient methods such as LoRA, which could further mitigate memory requirements if needed.
>
> **W3: About experiments on larger models**
>
> Due to resource limitations, we were unable to include evaluations for larger models such as LLaMA2-7B or LLaMA2-13B in this submission, but we plan to address this in future work.
>
> Regarding the ablation study, we recognize the importance of jointly analyzing Hessian and clipping. The value of the Hessian fluctuates significantly, making it challenging to use directly. Most algorithms employ smoothing techniques to address this issue-for example, HiZZO[1] applies EMA and square root, AdaHessian[2] uses spatial averaging, and Sophia[3] clips the Newton update $\mathbf{H}^{-1}\mathbf{g}$. The existing optimizers leveraging Hessians have rarely conducted detailed ablation studies to isolate the standalone impact of the Hessian. Thus, we  jointly analyzing Hessian and clipping.
> Figure 5a presents an analysis of HELENE's three key components:
> From the blue curve to the orange curve, adding the annealing mechanism eliminates EMA-induced bias, improving stability. HELENE (purple) incorporates clipping into the diagonal Hessian, which significantly reduces the number of steps required for convergence (3000 steps).These results demonstrate that each of the components contributes meaningfully to accelerating zeroth-order optimization.
>
> **W4: About typos and wording errors**
> Thanks you for pointing out the typos, we have corrected it.
>
> [1] Zhao, Yanjun, et, al. "Second-order fine-tuning without pain for llms: A hessian informed zeroth-order optimizer" arXiv preprint arXiv:2402.15173 (2024).
>
> [2] Yao, Zhewei, et, al. "ADAHESSIAN: An Adaptive Second Order Optimizer for Machine Learning" AAAI 2021.
>
> [3] Liu, Hong, et, al. "Sophia: A Scalable Stochastic Second-order Optimizer for Language Model Pre-training" ICLR 2024.

---

> ### Author Response · Authors · 2024-11-28
> **Response to Reviewer Yry6 (2/2)**
>
> We sincerely thank the reviewer for their insightful comments and the valuable references provided. We greatly appreciate the feedback and will carefully incorporate the suggestions and cited literature into our manuscript.
>
> ### Questions
> **Q1: About Zeroth-optimization and LoRA**
>
> We have addressed our memory usage above. For example, in OPT-1.3B, MeZO/zero-shot requires 4GB, ICL needs 6GB, Prefix Fine-Tuning uses 19GB, and full-parameter fine-tuning consumes 27GB, while HELENE requires only 14GB. Our optimizer is fully compatible with LoRA, our method is not focused on parameter-efficient tuning and should not be compared with LoRA. Our optimizer is compatible with Lora, if we choose to fine-tune OPT-1.3B with HELENE(Lora,rank=8), the memory usage is approximately 330MB, 2.3% of 14GB.
>
> **Q2: About training time**
>
> We provide our actual training time of Helene as well as MeZO in the following table. Helene shows faster convergence rates compared with MeZO.
> ||Running Time (20,000 step)|Steps to convergence|Time to convergence| 	Speed up compared to MeZO|
> |-----|-----|-----|-----|-----|
> |MeZO|95 minuetes|20,000|95 minutes|1x|
> |Helene|150 minutes|6,000|45 minutes|2.11x|
>
> **Q3: About the annealing mechanism**
>
> Thank you for pointing out the confusion. Unlike Gradient Descent methods, Zero-Order methods require a stronger emphasis on momentum due to the increasing noise in SPSA gradient estimation as optimization progresses, as shown in Appendix Figure 7. When the gradient is given more weight relative to past momentum, training becomes unstable and may even diverge. The observed increase in noise in SPSA likely stems from the greater difficulty in optimization as model parameters approach a local minimum, where the noise from sampling perturbations can surpass the signal from the true gradient.  To address this, we propose a novel annealing strategy specifically designed for Zero-Order methods, where the factor $\alpha$ dynamically reduce the gradient's contribution in the momentum update.
>
> **Q4: About Algorithm2**
>
> Thank you for the reviewer’s comments highlighting the unclear statement. We have updated the manuscript to include a step-by-step derivation for clarity.
>
> **Q5: For HELENE in first-order optimization**
>
> MeZO is a framework with zeroth-order optimization designed to reduce memory cost of fine-tuning LLMs. HELENE is a MeZO method estimating second-order information to accelerate convergence.
>
> As stated in our previous answer, HELENE introduces three key components for its framework: Diagonal Hessian Estimation, the Annealing Mechanism, and Layerwise Clipping. For first-order optimizers, Hessian information is not applicable, so we did not conduct experiments in this context. The effect of Annealing can be observed in Figure 5a, where the difference between the blue and orange lines highlights its impact on the first-order optimizer. To demonstrate the effect of Clipping, we performed an experiment on image classification tasks.
>
> We apply layerwise clipping on MeZO-SGD for Image classification task on the dataset CIFAR-10, the following result demonstrates the effectiveness of the layerwise clipping.
> |Batch Size|1000|2000|3000|
> |-----|-----|-----|-----|
> |Test Accuracy (clip)|87.71|87.3|86.4|
> |Test Accuracy (No layerwise clip)|86.31|86.1|85.9|

---

> > ### Comment · Reviewer_Yry6 · 2024-12-02
> >
> > Thank you for your response. After reviewing your clarifications, I will set my final rating as 3. I believe that additional experiments are necessary to better substantiate the claims made in the paper, and further improvements to the presentation are essential to enhance its clarity and coherence.

---

### Official Review · Reviewer_GGuK · 2024-11-04

**Soundness:** 3
**Presentation:** 3
**Contribution:** 2
**Rating:** 5
**Confidence:** 4

**Summary:**

This paper proposes HELENE, which builds upon prior work, MeZO, by integrating a second-order preconditioning method designed to achieve faster, more stable convergence while maintaining a low memory footprint. The authors evaluate HELENE on prominent models, RoBERTa-large and OPT-1.3B, and report promising results in terms of speedup and accuracy.

**Strengths:**

HELENE’s integration of annealed A-GNB gradients and layer-wise clipping shows the authors' awareness of the specific computational nuances of finetuning LLM architectures, the discussion of EMA showed the and other related discussion showed good motivation.

Plus, memory saving is always crucial for better finetuning LLMs these days, HELENE's layer-wise approach is a solid step to conserve memory.

**Weaknesses:**

In the core algorithm:
- Gradient Computation (Steps 4–5): HELENE computes the gradient based on Simultaneous Perturbation Stochastic Approximation (SPSA), a zeroth-order optimization technique, which allows it to approximate gradients without needing backpropagation. This step saves memory, which is important for large models. However, SPSA tends to converge more slowly than direct gradient methods, and while the authors use annealing, it’s unclear if this fully mitigates the slower convergence.
- Annealing Schedule (Step 6): The annealing parameter α=Anneal(t) in Step 6 adjusts the moving average coefficient based on iteration count. the improvement here compared to EMA is not obvious in figure 5? (is this a wrong figure link)
- Although layerwise clipping seems beneficial, it also introduces additional hyperparameters and tuning complexity.
- The authors assert that the inclusion of the Hessian diagonal significantly improves convergence rates, but diagonal Hessian approximation methods generally struggle to capture the full curvature dynamics in deep networks.
For HELENE to have a true advantage, empirical evidence comparing convergence rates with Adam and other optimizers is crucial:
the accuracy improvement of 1.5% might not fully justify the added implementation and tuning complexity, especially if simpler optimizers (like Adam or AdamW)

**Questions:**

The 20× speedup is largely theoretical here, as HELENE’s slower convergence due to SPSA may offset this benefit in practice.
could you kindly provide a detailed profiling log comparing the actual run time?

---

> ### Author Response · Authors · 2024-11-27
> **Response to Reviewer GGuK (1/2)**
>
> ### Weakness
> **W1：About Slower than direct gradient-based method**:
>
> HELENE, our proposed zero-order method, is designed to enhance convergence speed compared to other zero-order methods like MeZO and MeZO-Adam. This paper specifically focuses on zero-order methods for memory-efficient training of large models. Therefore, comparing HELENE with first-order (i.e., gradient-based) methods falls outside the scope of this paper. The purpose of the annealing mechanism is to mitigating the bias introduced by native gradient EMA. The result can be found in Figure5a.
>
> **W2：About annealing**
>
>
> We revised the figure to further clarify how ablation study was organized. The annealing mechanism mitigates the gradient noise as optimization progress. When approaching the local minimum, the magnitude of noise in gradient estimation using SPSA could be greater than the magnitude of gradient, which can not be addressed well by native gradient EMA. As shown in the ablation study in our revised Figure 5(a), the EMA with annealing mechanism (orange curve) maintains a lower loss as steps increase compared to the EMA (blue curve), effectively mitigating the noise influence in ZO optimization process.
>
> **W3: About additional hyperparameters**
>
> We acknowledge that layer-wise clipping introduces additional hyperparameters. However, Figure 6 demonstrates that these hyperparameters are not highly sensitive. To address concerns about added tuning complexity, we found that using a percentile-based criterion for each layer—selecting parameters based on their absolute values within a top percentage (e.g., top 10%)—yields good performance. This approach reduces the tuning process to a single key hyperparameter, the clipping ratio, thereby significantly simplifying the procedure.
>
> **W4: About performances and convergence rates**
> 1. In terms of training accuracy, Table 3, shown below, may address your concerns. Helene generally outperforms most Zero-Order optimizers or at least matches their performance, while also achieving significant speed improvements. As shown in Figure 3, our algorithm consistently accelerates different downstream tasks and parameter-efficient fine-tuning, demonstrating its robustness and efficiency.
> 2. Regarding the convergence rate, as illustrated in Figure 4, our algorithm consistently outperforms Adam, AdamW, Lion, and other optimizers in fine-tuning the OPT/1.3B model on the SST-2 dataset.
>
> | Method         | Roberta-Large (FT) | Roberta-Large (LoRA) | Roberta-Large (Prefix) | OPT-1.3B (FT) | OPT-1.3B (LoRA) | OPT-1.3B (Prefix) |
> |-----------------|--------------------|-----------------------|-------------------------|---------------|------------------|-------------------|
> | FO-SGD         | 91.4               | 91.2                  | 89.6                    | 91.1          | 93.6             | 93.1              |
> | Forward-Grad   | 90.1               | 89.7                  | 89.5                    | 90.3          | 90.3             | 90.0              |
> | ZO-SGD         | 89.4               | 90.8                  | 90.0                    | 90.8    |    90.1     | 91.4              |
> | ZO-SGD-MMT     | 89.6               | 90.9                  | 90.1                    | 85.2          | 91.3             | 91.2              |
> | ZO-SGD-Cons    | 89.6               | **91.6**    | 90.1                    | 88.3          | 90.5        | 81.8       |
> | ZO-SGD-Sign    | 52.5     | 90.2    | 53.6        | 87.2 | 91.5 | 89.5 |
> | ZO-Adam        | 89.8               | 89.5                  | 90.2                    | 84.4          | **92.3**         | 91.4              |
> | HELENE         | **92.6**     | 90.6      | **91.7**                | **90.8** | 91.4  | **92.4**  |
>
>
> We also report and compare the performances of MeZO-Adam and HELENE on RoBERTa-Large in the following table. Helene shows an improvement of 0.5% compared with MeZO-Adam.
>
> |  | SST-2 | SST-5 | SNLI | MNLI | RTE | TREC |
> | ------ | ------ | ------ | ------ | ------ | ------ | ------ |
> | MeZO-Adam | 90.4 | 45.4 | 74.1 | 64.3 | 59.2 | 78.3 |
> | Helene | 92.6 | 46.7 | 72.0 | 58.9 | 65.7 | 78.1 |

---

> ### Author Response · Authors · 2024-11-27
> **Response to Reviewer GGuK (2/2)**
>
> ### Question
>
> **Q1: About actual running time**
>
> To evaluate empirical convergence performance, we repeated the experiments using the LLM OPT-1.3B model on the SST-2 dataset with the same batch size and summarized the time costs and empirical convergence rates in the table below.
>
> ||Running Time (20,000 step)|Steps to convergence|Time to convergence| 	Speed up compared to MeZO|
> |-----|-----|-----|-----|-----|
> |MeZO|95 minuetes|20,000|95 minutes|1x|
> |Helene|150 minutes|6,000|45 minutes|2.11x|

---

> > ### Comment · Reviewer_GGuK · 2024-11-30
> > **Appreciate your effort.**
> >
> > I appreciate the authors answering my questions and ran additional experiments. The paper will be improved over the previous version if you update these info.
> > I am happy to raise my score to 5 (mainly as an ecnouragement), but I don't think this draft at the current stage meets the publishing bar at ICLR yet, I encourage the authors to work on the revision and go for the next venue.
> > Meanwhile, I suggest the authors take a slightly different angle looking at the PEFT community, even if Zeroth-order methods are more of a "fair" baseline to adopt, without convincing empirical results, not many people in the community would actually adopt zero-th order finetuning method. The default finetuning methods for average practitioners /researchers are still going to be AdamW. You need to pitch something that's appealing/easy to use for ppl to give it a shot first.

---

> > > ### Author Response · Authors · 2024-12-02
> > > **Response to Reviewer GGuK**
> > >
> > > Thank you for your reply and encouragement! Zeroth-order optimization has evolved into a mature and impactful field, originating with applications in black-box adversarial attacks and gradually expanding to fine-tuning large-scale models. Notably, MeZO, published at NeurIPS 2023, has already garnered over 150 citations, demonstrating its influence and rapid adoption. This foundational work has inspired extensions into other domains, such as DPZero (ICML 2024), ZO-AdaMU (AAAI 2024), and ZO-SVRG (ICML 2024).
> > >
> > > These advancements highlight the growing interest and versatility of zeroth-order methods. Thus, we think that zeroth-order optimizer has attracted the attention of more scholars and is considered effective.
> > >
> > > **Comparison Between Zeroth-Order and Parameter-Efficient Fine-Tuning (PEFT)**
> > >
> > > While the community focusing on PEFT is larger, zeroth-order optimization offers unique advantages, particularly in memory efficiency and enabling ***full-parameter*** fine-tuning while retaining all model parameters during training, which is **fundamentally different with LoRA**.
> > >
> > > A key benefit of zeroth-order methods, such as our optimizer Helene, is their memory efficiency. For example:
> > > Training an OPT-13B model (on the MultiRC dataset, 400 tokens per example on average) using first-order SGD consumes approximately 97 GB memory for full-parameter fine-tuning, while SGD-LoRA requires 69 GB.
> > > For AdamW, First-order AdamW demands 240 GB of memory for full-parameter fine-tuning, and AdamW-LoRA requires 70 GB.
> > >
> > > In contrast, zeroth-order Helene requires only 90 GB for full-parameter fine-tuning and just 35 GB when combined with LoRA.
> > >
> > > This demonstrates that while PEFT methods like LoRA significantly reduce memory consumption, **Helene (zeroth-order) can complement PEFT approaches and reduces memory usage further**. Additionally, Helene provides the flexibility to operate in both full-parameter and parameter-efficient modes, maintaining compatibility with LoRA and other PEFT techniques.
> > >
> > > **Empirical Results between First-order and Zeroth-order methods**
> > >
> > > We conducted experiments of fine-tuning using RoBERTa-large on the SST-2 dataset to evaluate the performances of different optimization methods. Specifically, we compared first-order optimizers (FO-SGD, FO-Adam, FO-AdamW) with their zeroth-order counterparts (ZO-SGD, ZO-Adam, ZO-AdamW) and Helene, our proposed zeroth-order optimizer.
> > >
> > > | Method         | Roberta-Large (FT) | Roberta-Large (LoRA) | Roberta-Large (Prefix) |
> > > |-----------------|--------------------|-----------------------|-------------------------|
> > > | FO-SGD         | 91.4               | 91.2                  | 89.6                    |
> > > | FO-Adam         |  91.9               | 91.4                  | **91.9**                     |
> > > | FO-AdamW         |  92.4               | **92.1**                  | 91.7                    |
> > > | ZO-SGD         | 89.4               | 90.8                  | 90.0                    |
> > > | ZO-Adam        | 89.8               | 89.5                  | 90.2                    |
> > > | ZO-AdamW        | 90.5               | 86.7                  | 89.8                    |
> > > | HELENE         | **92.6**     | 90.6      | 91.7          |
> > >
> > > These results demonstrate that Helene achieves performance comparable to first-order AdamW in terms of accuracy. However, Helene achieves these results with **significantly reduced GPU resource requirements**. This highlights the practical advantages of zeroth-order optimization, particularly in resource-constrained environments, without compromising on model performance.
> > >
> > > [1] Malladi, Sadhika, et, al. "Fine-tuning language models with just forward passes." Advances in Neural Information Processing Systems 36 (2023): 53038-53075.
> > >
> > > [2] Zhang, Liang, et, al. "DPZero: Private Fine-Tuning of Language Models without Backpropagation." In Forty-first International Conference on Machine Learning, 2024.
> > >
> > > [3] Jiang, Shuoran, et, al. "ZO-AdaMU Optimizer: Adapting Perturbation by the Momentum and Uncertainty in Zeroth-Order Optimization." In Proceedings of the AAAI Conference on Artificial Intelligence, 2024.

---

### Meta-Review · Area_Chair_2MNu · 2024-12-19

**Metareview:**

This paper proposes a method for convergence acceleration of MeZO, a very popular zeroth order LLM fine tuning method. The authors primarily add a biased (but controllable bias) diagonal Hessian approximation as a preconditioner. The paper also adds momentum annealing and layer-wise clipping. While I think the paper is well on its way, none of the reviewers ultimately recommend acceptance after the author feedback period and I think there are clear areas of improvement remaining, see below.

**Additional Comments On Reviewer Discussion:**

The paper has been extensively updated, leading to substantial new results that need to be included and reviewed. Beyond this, I think there are a few points raised by the reviewers that remain valid. Most notably to me is the added memory overhead here -- in their own experiments, the authors' method uses 14 GB compared to 4 GB for MeZO.

While this is clearly substantially less than full parameter fine tuning, it's not obvious that we're in the same ballpark as MeZO here to warrant MeZO being the primary point of comparison. Notably, for example, bold in Table 1 appears to be "performed better than MeZO alone," but note that FT (Prefix) apparently used 19 GB of memory, for results that vary between comparable and substantially better. While the fact that HELENE substantially closes the gap between MeZO and FT (Prefix), it probably needs more thoughtful discussion, and certainly the usage of bold in the table is debatable as effectively saying "we fully discount this method over 5 GB."

---

### Decision · Program_Chairs · 2025-01-22

Reject